# Online Improper Learning with an Approximation Oracle*

**Elad Hazan**
Princeton University & Google AI Princeton
ehazan@cs.princeton.edu

**Wei Hu**
Princeton University
huwei@cs.princeton.edu

**Yuanzhi Li**
Stanford University
yuanzhil@stanford.edu

**Zhiyuan Li**
Princeton University
zhiyuanli@cs.princeton.edu

## Abstract

We study the following question: given an efficient approximation algorithm for an optimization problem, can we learn efficiently in the same setting? We give a formal affirmative answer to this question in the form of a reduction from online learning to offline approximate optimization using an efficient algorithm that guarantees near optimal regret. The algorithm is efficient in terms of the number of oracle calls to a given approximation oracle – it makes only logarithmically many such calls per iteration. This resolves an open question by Kalai and Vempala, and by Garber. Furthermore, our result applies to the more general improper learning problems.

## 1 Introduction

A fundamental question in learning theory is whether one can *efficiently* learn a given problem using an optimization oracle. Namely, does efficient offline optimization for a certain problem imply efficient learning algorithm for the same setting?

For online learning in games, it was shown by Kalai and Vempala (2005) that an optimization oracle giving the best decision in hindsight is sufficient for attaining optimal regret. However, in many non-convex settings, such an optimization oracle is either unavailable or NP-hard to compute. In the face of NP-hardness, algorithm designers resort to approximation algorithms that are guaranteed to return a solution within a certain multiplicative factor of the optimum. We give numerous examples in Section 1.2.

Kakade et al. (2009) considered the question of whether such an approximation algorithm is sufficient to obtain vanishing regret compared with an approximation to the best solution in hindsight. They gave an algorithm for this offline-to-online conversion. However, their reduction is inefficient in the number of per-iteration queries to the approximation oracle, which grows linearly with time. Ideally, an efficient reduction should call the oracle only a constant number of times per iteration and guarantee optimal regret at the same time, and this was considered an open question in the literature.

Various authors have improved upon this original online-to-offline reduction under certain cases, as we survey below. Recently, Garber (2017) made significant progress by giving a more efficient reduction, which improves the number of oracle calls in both full information and bandit settings. He explicitly asked whether a near-optimal reduction with only logarithmically many calls per iteration exists.

## 1.1 Problem Setting and Our Results

In this paper we resolve this question on the positive side and in a more general setting, which we formally define now.

**Formal description of problem setting.** We consider the standard setting of online linear optimization which is known to generalize statistical learning (Hazan, 2016; Shalev-Shwartz, 2012). In a repeated game, in round $t$ a player chooses a point $x_t$ from a decision set $\mathcal{K} \subseteq \mathbb{R}^d$ while an adversary chooses a loss vector $f_t \in \mathbb{R}^d$, which determines the loss of the player $f_t^\top x_t$ in this round. The loss vector $f_t$ is revealed to the player *after* her choice $x_t$ is made. We sometimes treat $f_t$ as a function on $\mathbb{R}^d$, i.e., $f_t(x) := f_t^\top x$.

Since we consider computationally intractable problems like maximum cut or minimum-rank matrix completion, we assume that the player has access to an offline optimization oracle. This oracle may return a point which does not belong to the target set $\mathcal{K}^*$, but rather to a different set $\mathcal{K}$. For example, in matrix completion the oracle may return a low-trace-norm matrix rather than a low-rank matrix.

This notion is formally captured by an optimization oracle $\mathcal{O}_{\mathcal{K},\mathcal{K}^*}$. Given an input $v \in \mathbb{R}^d$, this oracle outputs a point $\mathcal{O}_{\mathcal{K},\mathcal{K}^*}(v) \in \mathcal{K}$ which dominates all points in $\mathcal{K}^*$ in the direction $v$, that is

$$v^\top \mathcal{O}_{\mathcal{K},\mathcal{K}^*}(v) \leq \min_{x^* \in \mathcal{K}^*} v^\top x^*.$$

The goal for the player is to minimize her *regret*, which is the difference between her cumulative loss and that of the best single decision (in $\mathcal{K}^*$) in hindsight:

$$\text{Reg}_{\mathcal{K},\mathcal{K}^*}(T) := \sum_{t=1}^{T} f_t^\top x_t - \min_{x^* \in \mathcal{K}^*} \sum_{t=1}^{T} f_t^\top x^*.$$

We remark that the above problem setting is similar in spirit to the notion of *improper learning*, where one is allowed to output a hypothesis not from the target set. Therefore, we view the problem setting described above as an *online* version of improper learning.

In the special case of $\mathcal{K}^* = \alpha \mathcal{K}$ ($\alpha > 1$), $\mathcal{O}_{\mathcal{K},\alpha\mathcal{K}}$ becomes an $\alpha$-approximation oracle on $\mathcal{K}$, and the setting and the notation of regret are the same with those studied in (Kakade et al., 2009; Garber, 2017), i.e., $\text{Reg}_{\mathcal{K},\alpha\mathcal{K}}(T) := \sum_{t=1}^{T} f_t^\top x_t - \alpha \min_{x \in \mathcal{K}} \sum_{t=1}^{T} f_t^\top x$. This is called the $\alpha$-*regret*.

**Our results.** In this setting, we give two different algorithms, one based on the online mirror descent (OMD) method and another based on the continuous multiplicative weight update (CMWU) algorithm. Both of them give nearly optimal regret as well as oracle efficiency, while applying to general loss vectors. Our results are summarized in Table 1 below. We present these two algorithms and their guarantees in Sections 3 and Appendix B.

| Algorithm | Regret over $T$ rounds | Oracle calls per round | Loss vectors |
|---|---|---|---|
| Kakade et al. (2009) | $O(\sqrt{T})$ | $O(T)$ | general |
| Garber (2017) | $O(\sqrt{T})$ | $\tilde{O}(\sqrt{T})$ | non-negative |
| **Alg. 1 (this paper)** | $\boldsymbol{O(\sqrt{T})}$ | $\boldsymbol{O(\log T)}$ | PNIP property (Def. 2.4) |
| **Alg. 6 (this paper)** | $\boldsymbol{\tilde{O}(\sqrt{T})}$ | $\boldsymbol{O(\log T)}$ | general |

Table 1: Summary of results in the full information setting. The $\tilde{O}$ notation hides constant and logarithmic factors.

| Algorithm | Regret over $T$ rounds | Oracle calls in $T$ rounds | Loss vectors |
|---|---|---|---|
| Kakade et al. (2009) | $O(T^{\frac{2}{3}})$ | $O(T^{\frac{4}{3}})$ | general |
| Garber (2017) | $O(T^{\frac{2}{3}})$ | $\tilde{O}(T)$ | non-negative |
| **Alg. 3 (this paper)** | $\boldsymbol{O(T^{\frac{2}{3}})}$ | $\boldsymbol{\tilde{O}(T^{\frac{2}{3}})}$ | non-negative |

Table 2: Summary of results in the bandit setting.

In addition to these two algorithms, we give an improved result in the bandit setting. In this more difficult setting, the player cannot observe $f_t$, but rather only the loss she has suffered, namely the

scalar $f_t^\top x_t$. We show how to extend our mirror descent-based algorithm to the bandit setting and obtain the same $O(T^{2/3})$ regret as in (Kakade et al., 2009; Garber, 2017), but with a significantly lower computational cost. See Table 2 for a comparison. We present our bandit result in Section 4.

## 1.2 Applications

The setting of online learning with approximation algorithms has been well studied since (Kalai and Vempala, 2005) with numerous applications.

For example, in the online max-cut problem, a learner iteratively predicts a cut over a set of vertices $V$, and afterwards the adjacency information for two vertices is revealed. The loss is zero or one, depending on whether the learner correctly predicted the connectivity of the two vertices. The offline version of this problem is NP-hard, but admits SDP-based approximation algorithms such as the famous 0.878-approximation by Goemans and Williamson (1995). Our results imply an online algorithm that can predict as accurate as the best 0.878-approximation to the maximum cut in hindsight, and calls the SDP relaxation only logarithmically many times per iteration.

Numerous other examples exist for combinatorial graph optimization problems such as the traveling salesman problem, sparsest graph cut, etc. Other applications include prominent machine learning problems whose offline optimization problem is NP-hard, for example, matrix completion and recommendation systems. The reader is referred to (Kakade et al., 2009; Garber, 2017) for more detailed exposition of applications.

## 1.3 Related Work

The reduction from online learning to offline approximation algorithms was already considered by Kalai and Vempala (2005). Their scheme, based on the follow-the-perturbed-leader (FTPL) algorithm, requires very strong approximation guarantee from the approximation oracle, namely, a fully polynomial time approximation scheme (FPTAS), and requires an approximation that improves with time. Balcan and Blum (2006) used the same approach in the context of mechanism design.

Kalai and Vempala (2005) also proposed a specialized reduction that works under certain conditions on the approximation oracle, satisfied by some known algorithms for problems such as MAX-CUT. Fujita et al. (2013) further gave more general reductions that apply to problems whose approximation algorithms are based on convex relaxations of mathematical programs. Their scheme is also based on the FTPL method.

Recent advancements on black-box online-to-offline reductions were made in (Kakade et al., 2009; Dudík et al., 2016; Garber, 2017). Hazan and Koren (2016) showed that efficient reductions are in general impossible, unless special structure is present. In the settings we consider this special structure is a linear cost function over the space.

Our algorithms fall into one of two templates. The first is the online mirror descent method, which is an adaptive version of the follow-the-regularized-leader (FTRL) algorithm. The second is the continuous multiplicative weight update method, which dates back to Cover's portfolio selection method (Cover, 1991) and Vovk's aggregating algorithm (Vovk, 1990). The reader is referred to the books (Cesa-Bianchi and Lugosi, 2006; Shalev-Shwartz, 2012; Hazan, 2016) for details and background on these prediction frameworks. We also make use of polynomial-time algorithms for sampling from log-concave distributions (Lovász and Vempala, 2007).

## 2 Preliminaries

For $x \in \mathbb{R}^d$ and $r > 0$, denote by $B(x, r)$ the Euclidean ball in $\mathbb{R}^d$ of radius $r$ centered at $x$. For $\mathcal{S}, \mathcal{S}' \subseteq \mathbb{R}^d$, $\beta \in \mathbb{R}$, $y \in \mathbb{R}^d$ and $A \in \mathbb{R}^{d' \times d}$, define $\mathcal{S} + \mathcal{S}' := \{x + x' : x \in \mathcal{S}, x' \in \mathcal{S}'\}$, $\beta \mathcal{S} := \{\beta x : x \in \mathcal{S}\}$, $x + \mathcal{S} := \{x + y : y \in \mathcal{S}\}$, and $A\mathcal{S} := \{Ax : x \in \mathcal{S}\}$. The convex hull of $\mathcal{S} \subseteq \mathbb{R}^d$ is denoted by $\mathrm{CH}(\mathcal{S})$. Denote by $\mathrm{Vol}(\mathcal{S})$ the volume (Lebesgue measure) of a set $\mathcal{S} \subseteq \mathbb{R}^d$. Denote by $\Delta^{k-1}$ the probability simplex in $\mathbb{R}^k$.

A set $\mathcal{C} \subseteq \mathbb{R}^d$ is called a *cone* if for any $\beta \geq 0$ we have $\beta \mathcal{C} \subseteq \mathcal{C}$. For any $\mathcal{S} \subseteq \mathbb{R}^d$, define the *dual cone* of $\mathcal{S}$ as $\mathcal{S}^\circ := \{y \in \mathbb{R}^d : x^\top y \geq 0, \ \forall x \in \mathcal{S}\}$. $\mathcal{S}^\circ$ is always a convex cone, even when $\mathcal{S}$ is

---

**Algorithm 1** Online Mirror Descent using a Projection-and-Separation Oracle

---

**Input:** Learning rate $\eta > 0$, tolerance $\epsilon > 0$, regularizer $\varphi$, convex cone $W$, time horizon $T \in \mathbb{N}_+$
  1: $y_1 \leftarrow \arg\min_{y \in \mathrm{Dom}(\varphi)} \varphi(y)$.
  2: **for** $t = 1$ **to** $T$ **do**
  3:     $(x_t, V = (v_1, \ldots, v_k), p) \leftarrow \mathcal{PAD}(y_t, \epsilon, W, \varphi)$
  4:     Play $\tilde{x}_t = v_i$ with probability $p_i$ $(i \in [k])$, and observe the loss vector $f_t$.
  5:     $\nabla\varphi(y_{t+1}) \leftarrow \nabla\varphi(x_t) - \eta f_t$
  6: **end for**

---

neither convex nor a cone. For any closed set $\mathcal{S} \subseteq \mathbb{R}^d$, define $\Pi_\mathcal{S} : \mathbb{R}^d \to \mathcal{S}$ to be the *projection* onto $\mathcal{S}$, namely $\Pi_\mathcal{S}(x) := \arg\min_{x' \in \mathcal{S}} \|x' - x\|^2$.

**Definition 2.1.** *A strictly convex function* $f : \mathcal{A} \to \mathbb{R}$ *($\mathcal{A} \subseteq \mathbb{R}^d$ is convex) is* Legendre *if* $\nabla f$ *is continuous in* $\mathrm{int}(\mathcal{A})$ *and for any sequence* $x_1, x_2, \cdots \in \mathcal{A}$ *converging to a boundary point of* $\mathcal{A}$, $\lim_{n \to \infty} \|\nabla f(x_n)\| = \infty$.

**Definition 2.2.** *For a Legendre function* $\varphi : \mathcal{A} \to \mathbb{R}$, *the* Bregman divergence *with respect to* $\varphi$ *is defined as* $D_\varphi(x, y) := \varphi(x) - \varphi(y) - \nabla\varphi(y)^\top(x - y)$ *($\forall x, y \in \mathcal{A}$).*

**Lemma 2.3** (Generalized Pythagorean theorem, see e.g. Lemma 11.3 in (Cesa-Bianchi and Lugosi, 2006)). *For any closed convex set* $\mathcal{S} \subseteq \mathbb{R}^d$, $x \in \mathbb{R}^d$, $y \in \mathcal{S}$, *and any Legendre function* $\varphi : \mathbb{R}^d \to \mathbb{R}$, *letting* $z = \arg\min_{x' \in \mathcal{S}} D_\varphi(x', x)$, *we must have* $D_\varphi(y, x) \geq D_\varphi(y, z) + D_\varphi(z, x)$.

**Definition 2.4** (Pairwise non-negative inner product). *For a twice-differentiable Legendre function* $\varphi : \mathcal{A} \to \mathbb{R}$ *with domain* $\mathcal{A} \subseteq \mathbb{R}^d$ *and a convex cone* $W \subseteq \mathbb{R}^d$, *we say* $(\varphi, W)$ *satisfies the* pairwise non-negative inner product (PNIP) *property, if for all* $w, w' \in W$ *and* $H \in \mathrm{CH}(\mathcal{H})$, *where* $\mathcal{H} = \{\nabla^2\varphi(x) : x \in \mathcal{A}\}$, *it holds that* $w^\top H^{-1} w' \geq 0$.

**Examples.** $(\varphi, W)$ satisfies the PNIP property if:

1. $\varphi(x) = \frac{1}{2}\|x\|^2$, $x \in \mathbb{R}^d$ and $W \subseteq W^\circ$, such as the non-negative orthant $\mathbb{R}^d_+$, the positive semidefinite matrix cone, and the Lorentz cone $L_{d+1} = \{(x, z) \in \mathbb{R}^d \times \mathbb{R} : \|x\|_2 \leq z\}$.

2. $\varphi(x) = \sum_{i=1}^d x_i(\log x_i - 1)$ (with domain $\mathbb{R}^d_+$) and $W = \mathbb{R}^d_+$;

3. $\varphi(x) = \frac{1}{2} x^\top Q^{-1} x$ (with domain $\mathbb{R}^d$), where $Q = MM^\top$, $M \in \mathbb{R}^{d \times d}$ is an invertible matrix, and $W = (M^\top)^{-1} \mathbb{R}^d_+$.

This is useful in our bandit algorithm in Section 4.

**Log-concave distributions.** A distribution over $\mathbb{R}^d$ with a density function $f$ is *log-concave* if $\log(f)$ is a concave function. For a convex set $\mathcal{S}$ equipped with a membership oracle, there exist polynomial-time algorithms for sampling from any log-concave distribution over $\mathcal{S}$ (Lovász and Vempala, 2007). This can be used to approximately compute the mean of any log-concave distribution. For ease of presentation, we will assume that we can compute the mean of bounded-supported log-concave distributions exactly. Detailed explanation is provided in Appendix D.

# 3 Mirror Descent with an Approximation Oracle

In this section, we give an efficient online improper linear optimization algorithm (Algorithm 1) in the full information setting based on online mirror descent (OMD) equipped with a strongly convex regularizer $\varphi$, which achieves $O(\sqrt{T})$ regret when the regularizer $\varphi$ and the domain of linear loss functions $W$ satisfy the *pairwise non-negative inner product (PNIP)* property (Definition 2.4).

We suppose $\mathcal{K}, \mathcal{K}^* \subseteq B(0, R)$, and the loss vectors $\{f_t\}$ come from a convex cone $W \subseteq \mathbb{R}^d$ and $\|f_t\| \leq L$ $(R, L > 0)$. Omitted proofs in this section are given in Appendix A.

**Theorem 3.1.** *Suppose* $(\varphi, W)$ *satisfies the PNIP property (Definition 2.4). Then for any* $\epsilon, \eta > 0$, *Algorithm 1 satisfies the following regret guarantee:*

$$\forall x^* \in \mathcal{K}^* : \quad \mathbb{E}\left[\sum_{t=1}^T (f_t(\tilde{x}_t) - f_t(x^*))\right] \leq \frac{1}{\eta}\left(\varphi(x^*) - \varphi(y_1) + \sum_{t=1}^T D_\varphi(x_t, y_{t+1})\right) + \epsilon L T.$$

In particular, if $\varphi$ is $\mu$-strongly convex and $A \geq \max_{x^* \in \mathcal{K}^*}(\varphi(x^*) - \varphi(y_1))$, setting $\epsilon = \frac{R}{T}$ and $\eta = \frac{1}{L}\sqrt{\frac{2\mu A}{T}}$, we have

$$\forall x^* \in \mathcal{K}^* : \quad \mathbb{E}\left[\sum_{t=1}^{T}(f_t(\tilde{x}_t) - f_t(x^*))\right] \leq L\sqrt{\frac{2AT}{\mu}} + LR,$$

and in this case, Algorithm 1 makes at most $\left\lceil 5d \log \left( \left( 6\sqrt{T} + \frac{4}{R}\sqrt{\frac{A}{\mu}} + 4 \right) T \right) \right\rceil$ calls of $\mathcal{O}_{\mathcal{K},\mathcal{K}^*}$ per round.

For the problem of $\alpha$-regret minimization using an $\alpha$-approximation oracle, we have the following regret guarantee, which is an immediate corollary of Theorem 3.1.

**Corollary 3.2.** *If $W \subseteq \mathbb{R}_+^d$, $\mathcal{K} \subseteq B(0,R)$, $\mathcal{K}^* = \alpha\mathcal{K}$, $\varphi(x) = \frac{1}{2}\|x\|^2$, setting $\epsilon = \frac{\alpha R}{T}$, $\eta = \frac{\alpha R}{L\sqrt{T}}$, Algorithm 1 has the following regret guarantee:*

$$\forall x^* \in \mathcal{K} : \quad \mathbb{E}\left[\sum_{t=1}^{T}f_t(\tilde{x}_t) - \alpha\sum_{t=1}^{T}f_t(x^*)\right] = \mathbb{E}\left[\sum_{t=1}^{T}f_t(\tilde{x}_t) - \sum_{t=1}^{T}f_t(\alpha x^*)\right] \leq \alpha LR(\sqrt{T}+1).$$

Algorithm 1 is a variant of the OMD algorithm that makes use of a *projection-and-decomposition (PAD)* oracle, defined as follows:

**Definition 3.3** (Projection-and-decomposition oracle)**.** *A projection-and-decomposition (PAD) oracle onto $\mathcal{K}^*$, $\mathcal{PAD}(y,\epsilon,W,\varphi)$, is defined as a procedure that given $y \in \mathbb{R}^d$, $\epsilon > 0$, a convex cone $W$ and a Legendre function $\varphi$ produces a tuple $(y', V, p)$, where $y' \in \mathbb{R}^d$, $V = (v_1, \ldots, v_k) \in \mathbb{R}^{d \times k}$ and $p = (p_1, \ldots, p_k)^\top \in \Delta^{k-1}$, such that:*

1. *$y'$ is "closer" to $\mathcal{K}^*$ than $y$ with respect to the Bregman divergence of $\varphi$ (and hence is an "infeasible projection"): $\forall x^* \in \mathcal{K}^*$, $D_\varphi(x^*, y') \leq D_\varphi(x^*, y)$;*

2. *$v_1, \ldots, v_k \in \mathcal{K}$, and $\sum_{i=1}^{k} p_i v_i$ is a point that "almost dominates" $y'$ in all directions in $W$. In other words, there exists $c \in W^\circ$ such that $\|\sum_{i=1}^{k} p_i v_i + c - y'\| \leq \epsilon$.*

The purpose of the PAD oracle is the following. Suppose the OMD algorithm tells us to play a point $y$. Since $y$ might not be in the feasible set $\mathcal{K}$, we can call the PAD oracle to find another point $y'$ as well as a distribution $p$ over points $v_1, \ldots, v_k \in \mathcal{K}$. The first property in Definition 3.3 is sufficient to ensure that playing $y'$ also gives low regret, and the second property further ensures that we have a distribution of points in $\mathcal{K}$ that suffers less loss than $y'$ for every possible loss function so we can play according to that distribution.

Assuming the availability of a PAD oracle, one can use a standard analysis of OMD to prove a regret bound for Algorithm 1 as in Theorem 3.1. The proof is given in Appendix A.

Next we show how to construct a PAD oracle using the optimization oracle $\mathcal{O}_{\mathcal{K},\mathcal{K}^*}$. Our construction is given in Algorithm 2. Theorem 3.4 gives its guarantee.

**Theorem 3.4.** *Suppose $(\varphi, W)$ satisfies PNIP condition (Definition 2.4) and $\varphi$ is $\mu$-strongly convex. Then for any $y \in \mathbb{R}^d$, $\epsilon \in (0, R]$, Algorithm 2 terminates in $\left\lceil 5d \log \frac{4R + 2\sqrt{2 \min_{x^* \in \mathcal{K}^*} D_\varphi(x^*,y)/\mu}}{\epsilon} \right\rceil$ iterations, and it correctly implements the projection-and-decomposition oracle $\mathcal{PAD}(y,\epsilon,W,\varphi)$, i.e., its output $(y', V, p)$ satisfies the two properties in Definition 3.3.*

**Remark.** *We can use random walk methods to compute an $\frac{1}{T}$-approximation of the gravity center (line 3 in Algorithm 2) in $\mathrm{poly}(T)$ time, which is enough for the purpose of bounding regret. We can also replace the center of gravity method with the ellipsoid method, or any other optimization method with a similar "optimization interface" (i.e., any method that is based on separation queries and guarantees similar bounds on the number of iterations required to find a feasible point), as pointed out by Garber (2017). Specifically, using the ellipsoid method, we can significantly reduce the computational complexity to depend only polynomially in $\log T$ (rather than $T$), at the cost of a slightly higher oracle complexity, namely $O(d^2 \log T)$ calls to the oracle per round. We choose center of gravity over other optimization methods only because it has the best oracle complexity, which is the main focus of this paper.*

---

**Algorithm 2** Projection-and-Decomposition Oracle, $\mathcal{PAD}(y, \epsilon, W, \varphi)$

---

**Input:** Point $y \in \mathbb{R}^d$, tolerance $\epsilon > 0$, convex cone $W$, regularizer $\varphi$,
**Output:** $(y', V, p)$, where $y' \in \mathbb{R}^d$, $V = (v_1, \ldots, v_k) \in \mathbb{R}^{d \times k}$ for some $k$ such that $v_i \in \mathcal{K}$
   $\quad (\forall i \in [k])$, and $p = (p_1, \ldots, p_k)^\top \in \Delta^{k-1}$
1: $W_1 \leftarrow W \cap B(0,1)$, $\quad z_1 \leftarrow y$, $\quad i \leftarrow 0$
2: **while** $i < 5d \log \frac{2(R + \|z_{i+1}\|)}{\epsilon}$ **do**
3: $\quad i \leftarrow i+1$, $\quad w_i \leftarrow \frac{\int_{W_i} w \, dw}{\text{Vol}(W_i)}$, $\quad v_i \leftarrow \mathcal{O}_{\mathcal{K}, \mathcal{K}^*}(w_i)$.
4: $\quad z_{i+1} \leftarrow \underset{z \in \mathbb{R}^d, w_i^\top (z - v_i) \geq 0}{\arg\min} D_\varphi(z, z_i)$, $\quad W_{i+1} \leftarrow W_i \cap \{w \in \mathbb{R}^d : w^\top(v_i - z_{i+1}) \geq 0\}$.
5: **end while**
6: $k \leftarrow i$ and solve $\underset{p \in \Delta^{k-1}, c \in W^\circ}{\min} \|\sum_{i=1}^k p_i v_i + c - z_{k+1}\|$ to get $p$
7: **return** $y' = z_{k+1}, V = (v_1, \ldots, v_k), p$

---

We break the proof of Theorem 3.4 into several lemmas.

**Lemma 3.5.** *If $(\varphi, W)$ satisfies the PNIP condition (Definition 2.4), then $z_1, \ldots, z_{k+1}$ computed in Algorithm 2 satisfy $z_{i+1} - z_i \in W^\circ$ for all $i \in [k]$.*

*Proof.* Since we have $z_{i+1} = \underset{z \in \mathbb{R}^d : w_i^\top(z - v_i) \geq 0}{\arg\min} D_\varphi(z, z_i)$, by the KKT condition, we have

$$0 = \frac{\partial}{\partial z} \left( D_\varphi(z, z_i) - \lambda w_i^\top (z - v_i) \right) \Big|_{z = z_{i+1}} = \nabla \varphi(z_{i+1}) - \nabla \varphi(z_i) - \lambda w_i$$

for some $\lambda \geq 0$. On the other hand, note that $\nabla \varphi(z_{i+1}) - \nabla \varphi(z_i) = \int_0^1 \nabla^2 \varphi(\gamma z_{i+1} + (1 - \gamma) z_i) \cdot (z_{i+1} - z_i) d\gamma = H(z_{i+1} - z_i)$, for some $H \in \text{CH}(\mathcal{H})$, where $\mathcal{H} = \{\nabla^2 \varphi(x) : x \in \text{Dom}(\varphi)\}$. Therefore, for all $w \in W$ we have $w^\top(z_{i+1} - z_i) = w^\top H^{-1} H(z_{i+1} - z_i) = \lambda w^\top H^{-1} w_i \geq 0$. This means $z_{i+1} - z_i \in W^\circ$. $\quad\square$

**Lemma 3.6.** *Under the setting of Theorem 3.4, Algorithm 2 terminates in at most $\left\lceil 5d \log \frac{4R + 2\sqrt{2 \min_{x^* \in \mathcal{K}^*} D_\varphi(x^*, y)/\mu}}{\epsilon} \right\rceil$ iterations.*

*Proof.* According to the algorithm, for each $i$, $z_{i+1}$ is the Bregman projection of $z_i$ onto a half-space containing $\mathcal{K}^*$, since the oracle $\mathcal{O}_{\mathcal{K}, \mathcal{K}^*}$ ensures $w_i^\top v_i \leq w_i^\top x^*$ for all $x^* \in \mathcal{K}^*$. Then by the generalized Pythagorean theorem (Lemma 2.3) we know $D_\varphi(x^*, z_{i+1}) \leq D_\varphi(x^*, z_i)$ for all $x^* \in \mathcal{K}^*$ and $i$. Therefore we have $D_\varphi(x^*, z_i) \leq D_\varphi(x^*, z_1) = D_\varphi(x^*, y)$ for all $x^* \in \mathcal{K}^*$ and $i$.

Let $P := \min_{x^* \in \mathcal{K}^*} D_\varphi(x^*, y)$. Then there exists $x^* \in \mathcal{K}^*$ such that $P = D_\varphi(x^*, y) \geq D_\varphi(x^*, z_i) \geq \frac{\mu}{2} \|x^* - z_i\|^2$ for all $i$, where the last inequality is due to the $\mu$-strong convexity of $\varphi$. This implies $\|z_i\| \leq \|x^*\| + \sqrt{\frac{2P}{\mu}} \leq R + \sqrt{\frac{2P}{\mu}}$ for all $i$. Therefore, when $i \geq 5d \log \frac{4R + 2\sqrt{2P/\mu}}{\epsilon}$, we must have $i \geq 5d \log \frac{2(R + \|z_{i+1}\|)}{\epsilon}$, which means the loop must have terminated at this time. $\quad\square$

**Lemma 3.7.** *Under Theorem 3.4's setting, $\forall w \in W$, $\|w\| = 1$, $\exists i \in [k]$, such that $w^\top(v_i - y') \leq \epsilon$.*

*Proof.* We assume for contradiction that there exists a unit vector $h \in W$ such that $\min_{i \in [k]} h^\top(v_i - y') > \epsilon$. Note that $\|v_i - y'\| \leq \|v_i\| + \|y'\| \leq R + \|y'\|$. Letting $r := \frac{\epsilon}{2(R + \|y'\|)}$, we have $\forall w \in \frac{h}{2} + (W \cap B(0, r)) : \quad \min_{i \in [k]} w^\top(v_i - y') > 0$.

Since $r \leq \frac{1}{2}$ for $\epsilon \leq R$, we have $\frac{h}{2} + (W \cap B(0, r)) \subseteq \frac{h}{2} + (W \cap B(0, 1/2)) \subseteq W \cap B(0, 1) = W_1$. By the algorithm, we know that for all $w \in W_1 \setminus W_{k+1}$, there exists $i \in [k]$ such that $w^\top(v_i - z_{i+1}) \leq 0$. Notice that from Lemma 3.5 we know $z_{j+1} - z_j \in W^\circ$ for all $j \in [k]$. Thus for all $w \in W_1 \setminus W_{k+1}$ there exists $i \in [k]$ such that $w^\top(v_i - y') = w^\top(v_i - z_{k+1}) \leq w^T(v_i - z_{i+1}) \leq 0$. In other words, we have $\forall w \in W_1 \setminus W_{k+1} : \quad \min_{i \in [k]} w^\top(v_i - y') \leq 0$.

Therefore, we must have $\frac{h}{2} + (W \cap B(0, r)) \subseteq W_{k+1}$. We also have $\mathrm{Vol}(W_{i+1}) \leq (1 - 1/(2e))\mathrm{Vol}(W_i)$ for each $i \in [k]$ from Lemma D.2, since $W_{i+1}$ is the intersection of $W_i$ with a half-space that does not contain $W_i$'s centroid $w_i$ in the interior. Then we have

$$\mathrm{Vol}(W_1) = \mathrm{Vol}(W \cap B(0, 1)) = r^{-d}\mathrm{Vol}(W \cap B(0, r)) \leq r^{-d}\mathrm{Vol}(W_{k+1})$$
$$\leq r^{-d}(1 - 1/(2e))^k\mathrm{Vol}(W_1) < \mathrm{Vol}(W_1),$$

where the last step is due to $k \geq 5d\log\frac{1}{r} = 5d\log\frac{2(R+\|y'\|)}{\epsilon} = 5d\log\frac{2(R+\|z_{k+1}\|)}{\epsilon}$, which is true according to the termination condition of the loop. Therefore we have a contradiction. $\qquad\square$

The following lemma is a more general version of Lemma 6 in (Garber, 2017).

**Lemma 3.8.** *Given $v_1, \ldots, v_k \in \mathbb{R}^d$, $\epsilon \geq 0$ and a convex cone $W \in \mathbb{R}^d$, for any $x \in \mathbb{R}^d$, the following two statements are equivalent:*

(A) *There exists $p = (p_1, \ldots, p_k)^\top \in \Delta^{k-1}$ and $c \in W^\circ$ such that $\|\sum_{i=1}^{k} p_i v_i + c - x\| \leq \epsilon$.*

(B) *For all $w \in W$, $\|w\| = 1$, there exists $i \in [k]$ such that $w^\top(v_i - x) \leq \epsilon$.*

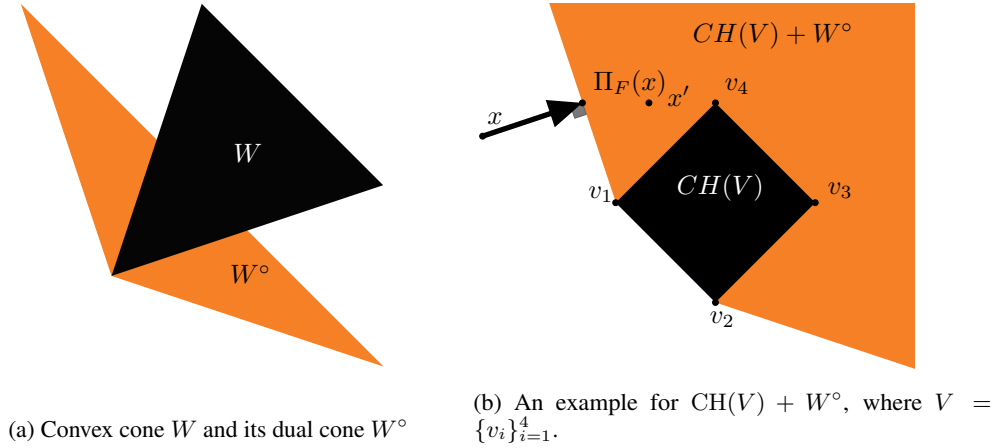

(a) Convex cone $W$ and its dual cone $W^\circ$

(b) An example for $CH(V) + W^\circ$, where $V = \{v_i\}_{i=1}^{4}$.

Figure 1: Geometric Interpretation of Lemma 3.8.

**Geometric interpretation of Lemma 3.8.** We defer the proof of Lemma 3.8 to Appendix A, and discuss its geometric intuition here. For simplicity of illustration, we only consider $\epsilon = 0$ here (Figure 1). First we look at the case where $W = \mathbb{R}^d$, $W^\circ = \{0\}$. In this case the lemma simply degenerated to the fact

$$x \in CH(\{v_i\}_{i=1}^{k}) \iff \text{There is no hyperplane that separates } x \text{ and all } v_i\text{'s.}$$

In the general case where $W \subseteq \mathbb{R}^d$ is an arbitrary convex cone, lemma 3.8 becomes

$$x \in CH(\{v_i\}_{i=1}^{k}) + W^\circ \iff \text{There is no direction } w \in W \text{ such that } w^\top x < w^\top v_i \text{ for all } i.$$

Denote $F := CH(\{v_i\}_{i=1}^{k}) + W^\circ$. For the "$\Rightarrow$" side, if $x \in F$, it is clear that for all $w \in W$ we must have $w^\top x \geq w^\top v_i$ for some $i$. For the "$\Leftarrow$" side, if $x \notin F$, then $w = \Pi_F(x) - x$ satisfies $w^\top x < w^\top v_i$ for all $i$. Moreover it is easy to see $\Pi_F(x) - x \in W$, which completes the proof.

Theorem 3.4 can be proved now using the above lemmas.

*Proof of Theorem 3.4.* The upper bound on the number of iterations is proved in Lemma 3.6. In the proof of Lemma 3.6, we have shown $D_\varphi(x^*, z_{i+1}) \leq D_\varphi(x^*, z_i)$ for all $x^* \in \mathcal{K}^*$ and $i$. This implies $D_\varphi(x^*, y') = D_\varphi(x^*, z_{k+1}) \leq D_\varphi(x^*, z_k) \leq \cdots \leq D_\varphi(x^*, z_1) = D_\varphi(x^*, y)$ for all $x^* \in \mathcal{K}^*$, which verifies the first property in Definition 3.3. The second property is a direct consequence of combining Lemmas 3.7 and 3.8. $\qquad\square$

---

**Algorithm 3** Online Stochastic Mirror Descent with Barycentric Regularization

---

**Input:** Learning rate $\eta > 0$, tolerance $\epsilon > 0$, $\{q_1, \ldots, q_d\}$ - a $\beta$-BS($\mathcal{K}$) for some $\beta > 0$, exploration probability $\gamma \in (0, 1)$, time horizon $T \in \mathbb{N}_+$

1: Instantiate Algorithm 1 with parameters $\eta, \epsilon, \varphi(x) = \frac{1}{2}x^\top Q^{-1}x$, $W' = (M^\top)^{-1}\mathbb{R}_+^d$, and $T$
2: **for** $t = 1$ **to** $T$ **do**
3:     Receive $\tilde{x}_t$ (the point to play in round $t$) from Algorithm 1
4:     $b_t \leftarrow \begin{cases} \text{EXPLORE}, & \text{with probability } \gamma \\ \text{EXPLOIT}, & \text{with probability } 1 - \gamma \end{cases}$
5:     **if** $b_t = \text{EXPLORE}$ **then**
6:         Sample $i_t \in [d]$ uniformly at random, and play $q_{i_t}$
7:         Receive loss $l_t = q_{i_t}^\top f_t$
8:         $\tilde{f}_t \leftarrow \frac{d}{\gamma}l_t Q^{-1}q_{i_t}$
9:     **else**
10:        Play $\tilde{x}_t$ and receive loss $l_t = \tilde{x}_t^\top f_t$
11:        $\tilde{f}_t \leftarrow 0$
12:     **end if**
13:     Feed $\tilde{f}_t$ to Algorithm 1 as the loss vector for round $t$ (Note that when $\tilde{f}_t = 0$, in the next round Algorithm 1 can simply play according to the distribution computed in this round without any oracle calls.)
14: **end for**

---

# 4 $\alpha$-Regret Minimization in the Bandit Setting

In this section we consider the $\alpha$-regret minimization problem in the bandit setting, where $W = \mathbb{R}_+^d$, $\mathcal{K} \subseteq \mathbb{R}_+^d \cap B(0, R)$ and $\mathcal{K}^* = \alpha\mathcal{K}$. Suppose the loss vectors $\{f_t\}$ come from $\mathbb{R}^d$ and $\|f_t\| \leq L$. Similar to (Kakade et al., 2009), we assume we know a $\beta$-*barycentric spanner* for $\mathcal{K}$. This concept was first introduced by Awerbuch and Kleinberg (2004).

**Definition 4.1** (Barycentric spanner). *A set of $d$ linearly independent vectors $\{q_1, \ldots, q_d\} \subset \mathbb{R}^d$ is a $\beta$-barycentric spanner for a set $\mathcal{K} \subset \mathbb{R}^d$, denoted by $\beta$-BS($\mathcal{K}$), if $\{q_1, \ldots, q_d\} \subseteq \mathcal{K}$ and for all $x \in \mathcal{K}$, there exist $\beta_1, \ldots, \beta_d \in [-\beta, \beta]$ such that $x = \sum_{i=1}^d \beta_i q_i$.*

Given $\{q_1, \ldots, q_d\}$ which is a $\beta$-BS($\mathcal{K}$), define $Q := \sum_{i=1}^d q_i q_i^\top$ and $M := (q_1, \ldots, q_d) \in \mathbb{R}^{d \times d}$.

**The need for a new regularization.** The bandit algorithm of Garber (2017) additionally requires a certain boundedness property of barycentric spanners, namely:

$$\max_{i \in [d]} q_i^\top Q^{-2} q_i \leq \chi.$$

However, for certain bounded sets this quantity may be unbounded, such as the two-dimensional axis-aligned rectangle with one axis being of size unity, and the other arbitrarily small. This unboundedness creates problems with the unbiased estimator of loss vector, whose variance can be as large as certain geometric properties of the decision set. To circumvent this issue, we design a new regularizer called *barycentric regularizer*, which gives rise to an unbiased estimator coupled with an online mirror descent variant that automatically ensures constant variance.

Similar to (Kakade et al., 2009; Garber, 2017), our bandit algorithm also simulates the full information algorithm with estimated loss vectors. Namely, our algorithm implements Algorithm 1 with a specific *barycentric regularizer* $\varphi(x) = \frac{1}{2}x^\top Q^{-1}x$. The algorithm is detailed in Algorithm 3, and its regret guarantee is given in Theorem 4.2. We prove Theorem 4.2 in Appendix C.

**Theorem 4.2.** *Denote by $z_t$ the point played by Algorithm 3 in round $t$.*

*Suppose we set $\eta = \frac{\alpha\beta^{4/3}}{LRT^{2/3}}$, $\epsilon = \frac{\alpha R}{T}$ and $\gamma = \frac{\beta^{2/3}d}{T^{1/3}}$ in Algorithm 3 (assuming $T > \beta^2 d^3$ so $\gamma < 1$). Then we have*

$$\forall x^* \in \mathcal{K}: \quad \mathbb{E}\left[\sum_{t=1}^T (f_t(z_t) - \alpha f_t(x^*))\right] \leq \alpha LR\left(3d(\beta T)^{2/3} + 1\right),$$

*and the expected total number of oracle calls to $\mathcal{O}_{\mathcal{K}, \alpha\mathcal{K}}$ in $T$ rounds is at most $O\left(d^2(\beta T)^{2/3}\log T\right)$.*

# 5 Conclusion and Open Problems

We have described two different algorithmic approaches to reducing regret minimization to offline approximation algorithms and maintaining optimal regret and poly-logarithmic oracle complexity per iteration, resolving previously stated open questions.

An intriguing open problem remaining is to find an efficient algorithm in the bandit setting that guarantees both $\tilde{O}(\sqrt{T})$ regret and $\text{poly}(\log T)$ oracle complexity per iteration (at least on average).

## Footnotes

*The full version of this paper can be found on https://arxiv.org/abs/1804.07837.

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
