[Supplementary Material]

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

*Proof.* First, for any fixed round $t \in [T]$, let $(x_t, V, p)$ be the output of $\mathcal{PAD}(y_t, \epsilon, W, \varphi)$ in this round. We know by the second property of the PAD oracle that there exists $c \in W^\circ$ such that $\|\sum_i p_i v_i + c - x_t\| \leq \epsilon$. Since $\tilde{x}_t$ is equal to $v_i$ with probability $p_i$, letting $\overline{x}_t := \mathbb{E}[\tilde{x}_t] = \sum_i p_i v_i$, we have

$$f_t(\overline{x}_t) - f_t(x_t) = \mathbb{E}[f_t(\tilde{x}_t) - f_t(x_t)] = f_t\left(\sum_i p_i v_i - x_t\right) \leq f_t\left(\sum_i p_i v_i - x_t + c\right) \leq \epsilon L. \tag{1}$$

We make use of the following properties of Bregman divergence, which can be verified easily (see e.g. Section 11.2 in (Cesa-Bianchi and Lugosi, 2006)):

$$\forall x, y, z : \quad (x - y)^\top(\nabla\varphi(z) - \nabla\varphi(y)) = D_\varphi(x, y) - D_\varphi(x, z) + D_\varphi(y, z). \tag{2}$$

Consider any $x^* \in \mathcal{K}^*$. We have

$$\sum_{t=1}^{T}(f_t(x_t) - f_t(x^*))$$

$$= \sum_{t=1}^{T}\frac{1}{\eta}\left(\nabla\varphi(x_t) - \nabla\varphi(y_{t+1})\right)^\top(x_t - x^*) \qquad \text{(by algorithm definition)}$$

$$= \frac{1}{\eta}\sum_{t=1}^{T}\left(D_\varphi(x^*, x_t) - D_\varphi(x^*, y_{t+1}) + D_\varphi(x_t, y_{t+1})\right) \qquad \text{(by (2))}$$

$$\leq \frac{1}{\eta}\sum_{t=1}^{T}\left(D_\varphi(x^*, y_t) - D_\varphi(x^*, y_{t+1}) + D_\varphi(x_t, y_{t+1})\right) \qquad \text{(by property of the PAD oracle)}$$

$$= \frac{1}{\eta}\left(D_\varphi(x^*, y_1) - D_\varphi(x^*, y_{T+1}) + \sum_{t=1}^{T} D_\varphi(x_t, y_{t+1})\right). \qquad \text{(by telescoping)}$$

$$\tag{3}$$

Combining (1) and (3), we can bound the expected improper regret of Algorithm 1 as

$$\forall x^* \in \mathcal{K}^* : \quad \mathbb{E}\left[\sum_{t=1}^{T}(f_t(\tilde{x}_t) - f_t(x^*))\right] = \sum_{t=1}^{T}(f_t(\overline{x}_t) - f_t(x^*))$$

$$\leq \frac{1}{\eta}\left(D_\varphi(x^*, y_1) - D_\varphi(x^*, y_{T+1}) + \sum_{t=1}^{T} D_\varphi(x_t, y_{t+1})\right) + \epsilon LT.$$

$$\tag{4}$$

By the optimality condition $\nabla\varphi(y_1)^\top(x^* - y_1) \geq 0$, we have

$$D_\varphi(x^*, y_1) \leq \varphi(x^*) - \varphi(y_1). \tag{5}$$

Plugging (5) into (4) and noting $D_\varphi(x^*, y_{T+1}) \geq 0$, we finish the proof of the first regret bound.

When $\varphi$ is $\mu$-strongly convex, for any fixed $y$, $f(x) = \varphi(x) - \nabla\varphi(y)^\top x$ is also strongly convex, achieving its unique minimum at $y$. From strong convexity we have

$$f(z) \geq f(x) + \nabla f(x)^\top(z - x) + \frac{\mu}{2}\|x - z\|^2, \ \forall x, z.$$

Minimizing both sides of the above inequality over $z$, we get $f(y) \geq f(x) - \frac{1}{2\mu}\|\nabla f(x)\|^2$. Thus,

$$D_\varphi(x, y) = f(x) - f(y) \leq \frac{1}{2\mu}\|\nabla f(x)\|^2 = \frac{1}{2\mu}\|\nabla\varphi(x) - \nabla\varphi(y)\|^2.$$

Then by the definition in Algorithm 1 we have

$$\forall t \in [T]: \quad D_\varphi(x_t, y_{t+1}) \leq \frac{1}{2\mu}\|\nabla\varphi(x_t) - \nabla\varphi(y_{t+1})\|^2 = \frac{1}{2\mu}\|\eta f_t\|^2 \leq \frac{\eta^2 L^2}{2\mu}. \tag{6}$$

From the above inequality and the choices of parameters $\epsilon = \frac{R}{T}$ and $\eta = \frac{1}{L}\sqrt{\frac{2\mu A}{T}}$, we have

$$\mathbb{E}\left[\sum_{t=1}^{T}(f_t(\tilde{x}_t) - f_t(x^*))\right] \leq \frac{A}{\eta} + \frac{\eta L^2 T}{2\mu} + LR \leq L\sqrt{\frac{2AT}{\mu}} + LR.$$

Next we bound the number of oracle calls made by Algorithm 1.

According to Theorem 3.4, round $t$ of Algorithm 1 calls $\mathcal{O}_{\mathcal{K}, \mathcal{K}^*}$ for at most

$$\left\lceil 5d \log \frac{4R + 2\sqrt{\frac{2}{\mu}\min_{x^* \in \mathcal{K}^*} D_\varphi(x^*, y_t)}}{\epsilon}\right\rceil$$

times. Hence it suffices to obtain an upper bound on $\min_{x^* \in \mathcal{K}^*} D_\varphi(x^*, y_t)$.

According to (4) (substituting $T$ with $t$), we have:

$$\forall t \in [T], \ \forall x^* \in \mathcal{K}^*: \quad D_\varphi(x^*, y_{t+1}) \leq D_\varphi(x^*, y_1) + \sum_{j=1}^{t} D_\varphi(x_j, y_{j+1}) - \eta\sum_{j=1}^{t}(f_j(\overline{x}_j) - f_j(x^*)) + \epsilon\eta Lt.$$

Plug (5) and (6) into the above inequality, we have

$$\forall t \in [T], \ \forall x^* \in \mathcal{K}^*: \quad D_\varphi(x^*, y_{t+1}) \leq D_\varphi(x^*, y_1) + \sum_{j=1}^{t} D_\varphi(x_j, y_{j+1}) - \eta\sum_{j=1}^{t}(f_j(\overline{x}_j) - f_j(x^*)) + \epsilon\eta Lt$$

$$\leq \varphi(x^*) - \varphi(y_1) + \frac{\eta^2 L^2}{2\mu}t - \eta\sum_{j=1}^{t}(f_j(\overline{x}_j) - f_j(x^*)) + \epsilon\eta Lt$$

$$\leq A + \frac{\eta^2 L^2}{2\mu}t + \eta\sum_{j=1}^{t}\|f_j\| \cdot \|\overline{x}_j - x^*\| + \epsilon\eta Lt$$

$$\leq A + \frac{\eta^2 L^2}{2\mu}T + 2\eta LRT + \epsilon\eta LT.$$

For $t = 1$ we also have $D_\varphi(x^*, y_1) \leq A$. Therefore Algorithm 1 calls $\mathcal{O}_{\mathcal{K}, \mathcal{K}^*}$ for at most

$$\left\lceil 5d \log \frac{4R + 2\sqrt{\frac{2}{\mu}\left(A + \left(\frac{\eta^2 L^2}{2\mu} + 2\eta LR + \epsilon\eta L\right)T\right)}}{\epsilon}\right\rceil$$

times per round.

When $\epsilon = \frac{R}{T}$ and $\eta = \frac{1}{L}\sqrt{\frac{2\mu A}{T}}$, we have

$$\frac{2}{\mu}\left(A + \left(\frac{\eta^2 L^2}{2\mu} + 2\eta LR + \epsilon\eta L\right)T\right) \leq \frac{4A}{\mu} + 6R\sqrt{\frac{2AT}{\mu}} \leq \left(2\sqrt{\frac{A}{\mu}} + 3R\sqrt{T}\right)^2,$$

so the number of oracle calls per iteration is at most $\left\lceil 5d\log\left(\left(6\sqrt{T} + \frac{4}{R}\sqrt{\frac{A}{\mu}} + 4\right)T\right)\right\rceil$. $\qquad\square$

Next we prove Lemma 3.8. First we need the following basic properties of projections.

**Lemma A.2** (Pythagorean theorem). *For any closed convex set $\mathcal{S} \subseteq \mathbb{R}^d$, $x \in \mathbb{R}^d$ and $y \in \mathcal{S}$, we have $(\Pi_{\mathcal{S}}(x) - x)^\top(\Pi_{\mathcal{S}}(x) - y) \leq 0$, or equivalently, $\|x - y\|^2 \geq \|\Pi_{\mathcal{S}}(x) - x\|^2 + \|\Pi_{\mathcal{S}}(x) - y\|^2$.*

**Lemma A.3.** *For any closed convex cone $W \subseteq \mathbb{R}^d$ and any $x \in \mathbb{R}^d$, we have $\Pi_W(x) - x \in W^\circ$.*

*Proof.* Since $W$ is a convex cone and $\Pi_W(x) \in W$, we have $w + \Pi_W(x) \in W$ ($\forall w \in W$). By Pythagorean theorem, we have $(\Pi_W(x) - x)^\top(y - \Pi_W(x)) \geq 0$ ($\forall y \in W$). Letting $y = w + \Pi_W(x) \in W$ ($\forall w \in W$), we get $(\Pi_W(x) - x)^\top w \geq 0$, which means $\Pi_W(x) - x \in W^\circ$. $\quad\square$

**Lemma A.4** (Restatement of Lemma 3.8). *Given $v_1, \ldots, v_k \in \mathbb{R}^d$, $\epsilon \geq 0$ and a convex cone $W \in \mathbb{R}^d$, for any $x \in \mathbb{R}^d$, the following two statements are equivalent:*

*(A) There exists $p = (p_1, \ldots, p_k)^\top \in \Delta^{k-1}$ and $c \in W^\circ$ such that $\|\sum_{i=1}^k p_i v_i + c - x\| \leq \epsilon$.*

*(B) For all $w \in W$, $\|w\| = 1$, there exists $i \in [k]$ such that $w^\top(v_i - x) \leq \epsilon$.*

*Proof.* Suppose (A) holds. Then for any $w \in W$, $\|w\| = 1$, we have

$$\min_{i\in[k]} w^\top(v_i - x) \leq w^\top\left(\sum_{i=1}^k p_i v_i - x\right) \leq w^\top\left(\sum_{i=1}^k p_i v_i + c - x\right) \leq \|w\| \cdot \left\|\sum_{i=1}^k p_i v_i + c - x\right\| \leq \epsilon.$$

So we have (A) $\implies$ (B).

Now suppose (B) holds. Let $(p^*, c^*) = \arg\min_{p\in\Delta^{k-1}, c\in W^\circ} \|\sum_{i=1}^k p_i v_i + c - x\|$. Since $0 \in W$, by the Pythagorean theorem (Lemma A.2), we have

$$\left\|\sum_{i=1}^k p_i^* v_i - x + c^*\right\| \geq \left\|\Pi_W\left(\sum_{i=1}^k p_i^* v_i - x + c^*\right)\right\|,$$

where the equality holds only when $\sum_{i=1}^k p_i^* v_i - x + c^* \in W$. Now we claim $\sum_{i=1}^k p_i^* v_i - x + c^* \in W$. Otherwise, letting $c' = c^* + \Pi_W\left(\sum_{i=1}^k p_i^* v_i - x + c^*\right) - \left(\sum_{i=1}^k p_i^* v_i - x + c^*\right)$, by Lemma A.3 we have $c' \in W^\circ$, and furthermore

$$\left\|\sum_{i=1}^k p_i^* v_i - x + c'\right\| = \left\|\Pi_W\left(\sum_{i=1}^k p_i^* v_i - x + c^*\right)\right\| < \left\|\sum_{i=1}^k p_i^* v_i - x + c^*\right\|,$$

which contradicts the optimality of $(p^*, c^*)$.

Thus we have $\sum_{i=1}^k p_i^* v_i + c^* - x \in W$. Let $w = \sum_{i=1}^k p_i^* v_i + c^* - x$ and $G = \text{CH}(\{v_1, \ldots, v_k\}) + W^\circ + \{-x\}$. Then we have $w = \Pi_G(0)$ by the definition of $(p^*, c^*)$. Since $G$ is convex and $v_i - x \in G$ for all $i \in [k]$, by the Pythagorean theorem (Lemma A.2) we have $w^\top(v_i - x - w) \geq 0$ for all $i \in [k]$, which implies $\|w\|^2 \leq \min_{i\in[k]} w^\top(v_i - x) \leq \epsilon\|w\|$, i.e., $\|w\| \leq \epsilon$. Hence we have (B) $\implies$ (A). $\quad\square$

# B Efficient Online Improper Linear Optimization via Continuous Multiplicative Weight Update (CMWU)

In this section, we design our second online improper linear optimization algorithm (in the full information setting) based on the continuous multiplicative weight update (CMWU) method. Same as in Section 3, we suppose $\mathcal{K}, \mathcal{K}^* \subseteq B(0, R)$, and the loss vectors $\{f_t\}$ come from a convex cone $W \subseteq \mathbb{R}^d$ and $\|f_t\| \leq L$ $(R, L > 0)$. Compared with Algorithm 1, the CMWU-based algorithm allows loss vectors to come from a general convex cone $W$ and does not require the PNIP condition (Definition 2.4).

## B.1 Separation-or-Decomposition Oracle

We first construct a *separation-or-decomposition (SOD)* oracle (Algorithm 4) using $\mathcal{O}_{\mathcal{K},\mathcal{K}^*}$, which we will use to design the online improper linear optimization algorithm later in this section. Given a point $x \in B(0, R)$, the SOD oracle either outputs a *separating hyperplane* between $x$ and $\mathcal{K}^*$, or outputs a *distribution of points in* $\mathcal{K}$ which approximately dominates $x$ in every direction in $W$. The guarantee of the SOD oracle is summarized in Theorem B.1.

---

**Algorithm 4** Separation-or-Decomposition Oracle, $\mathcal{SOD}(x, \epsilon, W)$

---

**Input:** Point $x \in B(0, R)$, tolerance $\epsilon > 0$, convex cone $W \subseteq \mathbb{R}^d$
**Output:** Decomposition $V = (v_1, \ldots, v_k) \in \mathbb{R}^{d \times k}$, $p = (p_1, \ldots, p_k)^\top \in \Delta^{k-1}$, such that $v_i \in \mathcal{K}$
  ($\forall i \in [k]$) and $\|\sum_{i=1}^k p_i v_i - x + c\| \leq 3\epsilon$ for some $c \in W^\circ$.
  **Or:** Separating hyperplane $(w, b) \in \mathbb{R}^d \times \mathbb{R}$, such that $\|w\| = 1$ and $w^\top x \leq b - \epsilon \leq \min_{x^* \in K^*} w^\top x^* - \epsilon$.
1: $k \leftarrow \lceil 5d \log \frac{4R}{\epsilon} \rceil$
2: $W_1 \leftarrow W \cap B(0, 1)$
3: **for** $i = 1$ **to** $k$ **do**
4:     $w_i \leftarrow \frac{\int_{W_i} w \, \mathrm{d}w}{\mathrm{Vol}(W_i)}$
5:     $v_i \leftarrow \mathcal{O}_{\mathcal{K},\mathcal{K}^*}(w_i)$
6:     **if** $w_i^\top x \leq w_i^\top v_i - \epsilon$ **then**
7:         **return** Separating hyperplane $\left( \frac{w_i}{\|w_i\|}, \frac{w_i^\top v_i}{\|w_i\|} \right)$
8:     **else**
9:         $W_{i+1} \leftarrow W_i \cap \{w \in \mathbb{R}^d : w^\top(v_i - x) \geq \epsilon\}$
10:     **end if**
11: **end for**
12: Solve $\min_{p \in \Delta^{k-1}, c \in W^\circ} \|\sum_{i=1}^k p_i v_i + c - x\|$ to get $p$
13: **return** $V = (v_1, \ldots, v_k), p$

---

**Theorem B.1.** *For any $x \in B(0, R)$ and $\epsilon \in (0, 2R]$, the separation-or-decomposition oracle in Algorithm 4, $\mathcal{SOD}(x, \epsilon, W)$, returns one of the two followings, using at most $k = \lceil 5d \log \frac{4R}{\epsilon} \rceil$ calls of $\mathcal{O}_{\mathcal{K},\mathcal{K}^*}$:*

1. *a decomposition $V = (v_1, \ldots, v_k) \in \mathbb{R}^{d \times k}$, $p = (p_1, \ldots, p_k)^\top \in \Delta^{k-1}$, such that $v_i \in \mathcal{K}$ ($\forall i \in [k]$) and $\|\sum_{i=1}^k p_i v_i - x + c\| \leq 3\epsilon$ for some $c \in W^\circ$.*

2. *a separating hyperplane $(w, b) \in \mathbb{R}^d \times \mathbb{R}$, where $\|w\| = 1$ and $w^\top x \leq b - \epsilon \leq \min_{x^* \in \mathcal{K}^*} w^\top x^* - \epsilon$.*

Before proving Theorem B.1, we first prove the following lemma using a similar argument in Lemma 3.7.

**Lemma B.2.** *For $x \in B(0, R)$ and $\epsilon \in (0, 2R]$, if $\mathcal{SOD}(x, \epsilon, W)$ returns a decomposition $(V, p)$, then for all unit vector $w \in W$, we have $\min_{i \in [k]} w^\top(v_i - x) \leq 3\epsilon$.*

*Proof.* Suppose that there exists a unit vector $h \in W$ such that $\min_{i \in [k]} h^\top (v_i - x) > 3\epsilon$. Note that $\|v_i - x\| \leq \|v_i\| + \|x\| \leq 2R$. Denoting $r = \frac{\epsilon}{4R}$, we have

$$\forall h' \in \frac{h}{2} + (W \cap B(0, r)) : \quad \min_{i \in [k]} h'^\top (v_i - x) > \epsilon.$$

Since $r \leq \frac{1}{2}$ for $\epsilon \leq 2R$, we have $\frac{h}{2} + (W \cap B(0, r)) \subseteq \frac{h}{2} + (W \cap B(0, 1/2)) \subseteq W \cap B(0, 1) = W_1$. Because the algorithm returns a decomposition, we have that after the last iteration,

$$\forall w \in W_1 \setminus W_{k+1} : \quad \exists i \in [k], \text{ s.t. } w^\top (v_i - x) \leq \epsilon.$$

Therefore, we must have $\frac{h}{2} + (W \cap B(0, r)) \subseteq W_{k+1}$. We also have $\mathrm{Vol}(W_{i+1}) \leq (1 - 1/2e)\mathrm{Vol}(W_i)$ from Lemma D.2 since $W_{i+1}$ is the intersection of $W_i$ with a half-space that does not contain $W_i$'s centroid. Then we have

$$\mathrm{Vol}(W_1) = \mathrm{Vol}(W \cap B(0, 1)) = r^{-d}\mathrm{Vol}(W \cap B(0, r)) \leq r^{-d}\mathrm{Vol}(W_{k+1})$$
$$\leq r^{-d}(1 - 1/2e)^k \mathrm{Vol}(W_1) < \mathrm{Vol}(W_1),$$

where the last step is due to $k \geq 5d \log \frac{1}{r} = 5d \log \frac{4R}{\epsilon}$. Therefore we have a contradiction. $\quad\square$

*Proof of Theorem B.1.* If $\mathcal{SOD}(x, \epsilon, W)$ returns a decomposition $(V, p)$, by Lemmas B.2 and 3.8, we know that there exists $c \in W^\circ$ such that $\|\sum_{i=1}^k p_i v_i + c - x\| \leq 3\epsilon$.

If $\mathcal{SOD}(x, \epsilon, W)$ returns a separating hyperplane $(w, b)$ at iteration $i \in [k]$, we know $w_i^\top x \leq w_i^\top v_i - \epsilon$. Since $w = \frac{w_i}{\|w_i\|}$, $b = w^\top v_i$ and $\|w_i\| \leq 1$, we have $w^\top x \leq w^\top v_i - \frac{\epsilon}{\|w_i\|} \leq b - \epsilon$. By the guarantee of $\mathcal{O}_{\mathcal{K},\mathcal{K}^*}$, we have $b - \epsilon = w^\top v_i - \epsilon \leq \min_{x^* \in K^*} w^\top x^* - \epsilon$.

The number of calls of $\mathcal{O}_{\mathcal{K},\mathcal{K}^*}$ is clearly upper bounded by $k = \lceil 5d \log \frac{4R}{\epsilon} \rceil$ since there are at most $k$ iterations and each iteration only calls $\mathcal{O}_{\mathcal{K},\mathcal{K}^*}$ once. $\quad\square$

## B.2 CMWU with Refining Domains

Now we look at a general online learning setting where the feasible domain is shrinking over time while being a superset of the target domain. Namely, suppose $\mathcal{K}^*$ is the target domain and $\mathcal{K}_t$ is the feasible domain in the $t$-th round. We assume $B(0, R) \supseteq \mathcal{K}_0 \supseteq \mathcal{K}_1 \supseteq \mathcal{K}_2 \cdots \supseteq \mathcal{K}_T \supseteq (1 - \gamma)\mathcal{K}^* + \gamma\mathcal{K}_0$ for some $\gamma \in (0, 1]$. In round $t$, the player only knows $\mathcal{K}_1, \ldots, \mathcal{K}_t$ and does not know $\mathcal{K}_j$ for all $j > t$. We can still run CMWU in this setting, using the knowledge of $\mathcal{K}_t$ at iteration $t$ - the algorithm is given in Algorithm 5. Theorem B.3 bounds the regret of Algorithm 5 in this setting.

---

**Algorithm 5** Continuous Multiplicative Weight Update (CMWU) with Refining Domains

---

**Input:** Learning rate $\eta > 0$, time horizon $T \in \mathbb{N}_+$
1: **for** $t = 1$ to $T$ **do**
2: $\quad$ Receive current domain $\mathcal{K}_t$
3: $\quad$ Play $x_t = \frac{\int_{\mathcal{K}_t} e^{-\eta \sum_{i=1}^{t-1} f_i(x)} x \mathrm{d}x}{\int_{\mathcal{K}_t} e^{-\eta \sum_{i=1}^{t-1} f_i(x)} \mathrm{d}x}$
4: $\quad$ Receive loss vector $f_t$
5: **end for**

---

**Theorem B.3.** *Suppose $B(0, R) \supseteq \mathcal{K}_0 \supseteq \mathcal{K}_1 \supseteq \mathcal{K}_2 \supseteq \cdots \supseteq \mathcal{K}_T \supseteq (1 - \gamma)\mathcal{K}^* + \gamma\mathcal{K}_0$ for $\gamma \in (0, 1]$. Then for any $0 < \eta \leq \frac{1}{LR}$, Algorithm 5 has the following regret guarantee:*

$$\forall x^* \in \mathcal{K}^* : \quad \sum_{t=1}^T (f_t(x_t) - f_t(x^*)) \leq \frac{d \log \frac{1}{\gamma}}{\eta} + \eta L^2 R^2 T + \gamma LRT - \frac{\sum_{t=1}^T \delta_t}{\eta},$$

*where*

$$\delta_t := \log \frac{\int_{\mathcal{K}_{t-1}} e^{-\eta \sum_{i=1}^{t-1} f_i(x)} \mathrm{d}x}{\int_{\mathcal{K}_t} e^{-\eta \sum_{i=1}^{t-1} f_i(x)} \mathrm{d}x}.$$

*In particular, setting $\gamma = 1/T$ and $\eta = \frac{1}{LR} \min\left\{1, \sqrt{\frac{d \log T}{T}}\right\}$, we have*

$$\forall x^* \in \mathcal{K}^* : \quad \sum_{t=1}^{T}(f_t(x_t) - f_t(x^*)) \leq LR\left(1 + 2\max\left\{\sqrt{dT \log T}, d \log T\right\}\right).$$

*Proof.* We fix any $x^* \in \mathcal{K}^*$ and denote $\bar{\mathcal{K}} := (1 - \gamma)x^* + \gamma \mathcal{K}_0$. Since $(1 - \gamma)\mathcal{K}^* + \gamma \mathcal{K}_0 \subseteq \mathcal{K}_T$, we have $\bar{\mathcal{K}} \subseteq \mathcal{K}_T$. We define

$$z_t(x) := e^{-\eta \sum_{i=1}^{t-1} f_i(x)}, \quad Z_t := \int_{\mathcal{K}_t} z_t(x) \mathrm{d}x, \quad Z_t' := \int_{\mathcal{K}_{t-1}} z_t(x) \mathrm{d}x.$$

A straightforward calculation gives us:

$$
\begin{aligned}
\log \frac{Z_{T+1}'}{Z_1'} &= \log\left(\frac{\int_{\mathcal{K}_T} e^{-\eta \sum_{t=1}^{T} f_t(x)} \mathrm{d}x}{\int_{\mathcal{K}_0} 1 \mathrm{d}x}\right) \\
&\geq \log\left(\frac{\int_{\bar{\mathcal{K}}} e^{-\eta \sum_{t=1}^{T} f_t(x)} \mathrm{d}x}{\int_{\mathcal{K}_0} 1 \mathrm{d}x}\right) \\
&= \log\left(\frac{\int_{\mathcal{K}_0} e^{-\eta \sum_{t=1}^{T} f_t((1-\gamma)x^* + \gamma x)} \gamma^d \mathrm{d}x}{\int_{\mathcal{K}_0} 1 \mathrm{d}x}\right) \\
&= \log\left(\frac{\int_{\mathcal{K}_0} e^{-\eta \sum_{t=1}^{T}((1-\gamma)f_t(x^*) + \gamma f_t(x))} \gamma^d \mathrm{d}x}{\int_{\mathcal{K}_0} 1 \mathrm{d}x}\right) \\
&\geq \log\left(\frac{\int_{\mathcal{K}_0} e^{-\eta \sum_{t=1}^{T}(f_t(x^*) + \gamma LR)} \gamma^d \mathrm{d}x}{\int_{\mathcal{K}_0} 1 \mathrm{d}x}\right) \\
&= d \log \gamma - \eta \sum_{t=1}^{T} f_t(x^*) - \eta\gamma LRT.
\end{aligned}
$$

On the other hand, we have

$$
\begin{aligned}
\log \frac{Z_{t+1}'}{Z_t} &= \log\left(\int_{\mathcal{K}_t} \frac{z_t(x)}{Z_t} e^{-\eta f_t(x)} \mathrm{d}x\right) \\
&\leq \log\left(\int_{\mathcal{K}_t} \frac{z_t(x)}{Z_t}\left(1 - \eta f_t(x) + (\eta f_t(x))^2\right) \mathrm{d}x\right) \\
&\leq \left(\int_{\mathcal{K}_t} \frac{z_t(x)}{Z_t}\left(1 - \eta f_t(x) + (\eta f_t(x))^2\right) \mathrm{d}x\right) - 1 \\
&\leq \int_{\mathcal{K}_t} \frac{z_t(x)}{Z_t}\left(-\eta f_t(x) + \eta^2 L^2 R^2\right) dx \\
&= -\eta f_t\left(\int_{\mathcal{K}_t} \frac{z_t(x)}{Z_t} x \mathrm{d}x\right) + \eta^2 L^2 R^2 \\
&= -\eta f_t(x_t) + \eta^2 L^2 R^2,
\end{aligned}
$$

where the first inequality is due to $e^a \leq 1 + a + a^2$ ($\forall a \leq 1$) and $|\eta f_t(x)| \leq \eta LR \leq 1$, the second inequality is due to $\log a \leq a - 1$ ($\forall a > 0$), and the third inequality is due to $|\eta f_t(x)| \leq \eta LR$.

Note that $\delta_t = \log \frac{Z_t'}{Z_t}$. Combining the two bounds above, we get:

$$
\begin{aligned}
\sum_{t=1}^{T}\left(-\eta f_t(x_t) + \eta^2 L^2 R^2\right) &\geq \sum_{t=1}^{T} \log \frac{Z_{t+1}'}{Z_t} = \log \frac{Z_{T+1}'}{Z_1'} + \sum_{t=1}^{T} \log \frac{Z_t'}{Z_t} \\
&\geq d \log \gamma - \eta \sum_{t=1}^{T} f_t(x^*) - \eta\gamma LRT + \sum_{t=1}^{T} \delta_t.
\end{aligned}
$$

In other words,

$$\sum_{t=1}^{T}(f_t(x_t) - f_t(x^*)) \leq \frac{d \log \frac{1}{\gamma}}{\eta} + \eta L^2 R^2 T + \gamma LRT - \frac{\sum_{t=1}^{T} \delta_t}{\eta}.$$

The second regret bound in the theorem follows directly by plugging in the values of $\gamma$ and $\eta$ stated in the theorem. Note that $\delta_t \geq 0$ for all $t \in [T]$ since $\mathcal{K}_t \subseteq \mathcal{K}_{t-1}$. $\qquad \square$

### B.3 Online Improper Linear Optimization via CMWU

Now we are ready to present our CMWU-based online improper learning algorithm. At a high level, this algorithm is a specialized implementation of Algorithm 5 for the online improper linear optimization problem. The algorithm starts with an initial convex domain $\mathcal{K}_0$ which is a superset of $\mathcal{K}^*$, and maintains a convex domain $\mathcal{K}_t$ at iteration $t$. In iteration $t$, the algorithm first computes the mean $x_t$ of a log-linear distribution over $\mathcal{K}_t$ using random walk, as in Line 3 of Algorithm 5. Then the algorithm calls the SOD oracle on $x_t$. If the SOD oracle returns a distribution of points in $\mathcal{K}$, then we can play according to that distribution, since the SOD oracle ensures that the expected loss of this distribution is not much larger than that of $x_t$. If the SOD oracle returns a separating hyperplane between $x_t$ and $\mathcal{K}^*$, then the algorithm replaces $\mathcal{K}_t$ with the intersection of the original $\mathcal{K}_t$ and the half-space given by this hyperplane that contains $\mathcal{K}^*$, and repeats the same process for the new $\mathcal{K}_t$ until a decomposition is returned by the SOD oracle. Note that each time $\mathcal{K}_t$ is updated, the mean $x_t$ of the log-linear distribution is *not* in the new $\mathcal{K}_t$, which according to Lemma D.2 implies that a constant probability mass is removed. This allows us to bound the total number of oracle calls. We detail our algorithm in Algorithm 6 and its regret bound in Theorem B.4.

---

**Algorithm 6** CMWU for Online Improper Linear Optimization

---

**Input:** Learning rate $\eta > 0$, tolerance $\gamma > 0$, initial convex domain $\mathcal{K}_0$, convex cone $W$, time horizon $T \in \mathbb{N}_+$

1: $\mathcal{K}_1 \leftarrow \mathcal{K}_0$
2: **for** $t = 1$ to $T$ **do**
3: $\quad x_t \leftarrow \frac{\int_{\mathcal{K}_t} e^{-\eta \sum_{i=1}^{t-1} f_i(x)} x \, \mathrm{d}x}{\int_{\mathcal{K}_t} e^{-\eta \sum_{i=1}^{t-1} f_i(x)} \, \mathrm{d}x}$
4: $\quad$ **while** $\mathcal{SOD}(x_t, 2\gamma R, W)$ returns a separating hyperplane $(w, b) \in \mathbb{R}^d \times \mathbb{R}$ **do**
5: $\quad\quad \mathcal{K}_t \leftarrow \mathcal{K}_t \cap \{x \in \mathbb{R}^d : w^\top x \geq b - 2\gamma R\}$
6: $\quad\quad x_t \leftarrow \frac{\int_{\mathcal{K}_t} e^{-\eta \sum_{i=1}^{t-1} f_i(x)} x \, \mathrm{d}x}{\int_{\mathcal{K}_t} e^{-\eta \sum_{i=1}^{t-1} f_i(x)} \, \mathrm{d}x}$
7: $\quad$ **end while**
8: $\quad$ Let $(V, p) \in \mathbb{R}^{d \times k} \times \Delta^{k-1}$ be the output of $\mathcal{SOD}(x_t, 2\gamma R, W)$
9: $\quad$ Play $\tilde{x}_t = v_i$ with probability $p_i$ $(i = 1, \ldots, k)$, where $V = (v_1, \ldots, v_k)$
10: $\quad \mathcal{K}_{t+1} \leftarrow \mathcal{K}_t$
11: $\quad$ Receive loss vector $f_t$
12: **end for**

---

**Theorem B.4.** *Suppose that the initial convex domain $\mathcal{K}_0$ satisfies $\mathcal{K}^* \subseteq \mathcal{K}_0 \subseteq B(0, R)$. Then for any $\gamma \in (0, 1]$ and $\eta \in \left(0, \frac{1}{LR}\right]$, Algorithm 6 satisfies the following regret guarantee:*

$$\forall x^* \in \mathcal{K}^* : \quad \mathbb{E}\left[\sum_{t=1}^{T}(f_t(\tilde{x}_t) - f_t(x^*))\right] \leq \frac{d \log \frac{1}{\gamma}}{\eta} + \eta L^2 R^2 T + 7\gamma LRT - \frac{s}{5\eta},$$

*where $s = \sum_{t=1}^{T} s_t$, and $s_t$ is the number of separating hyperplanes returned by the SOD oracle during round $t$.*

*In particular, if we set $\gamma = \frac{1}{T}, \eta = \frac{1}{LR} \min\left\{1, \sqrt{\frac{d \log T}{T}}\right\}$, then we have*

$$\forall x^* \in \mathcal{K}^* : \quad \mathbb{E}\left[\sum_{t=1}^{T}(f_t(\tilde{x}_t) - f_t(x^*))\right] \leq LR\left(7 + 2\max\left\{\sqrt{dT \log T}, d \log T\right\}\right),$$

*and in this case Algorithm 6 calls $\mathcal{O}_{\mathcal{K}, \mathcal{K}^*}$ for $O(dT \log T)$ times in $T$ rounds.*

*Proof.* In the proof, we use $\bar{\mathcal{K}}_t$ and $\bar{x}_t$ to denote the values of $\mathcal{K}_t$ and $x_t$ *at the end of iteration $t$* $(\bar{\mathcal{K}}_0 = \mathcal{K}_0)$. We define

$$z_t(x) := e^{-\eta \sum_{i=1}^{t-1} f_i(x)}, \quad Z_t := \int_{\bar{\mathcal{K}}_t} z_t(x)\mathrm{d}x, \quad Z'_t := \int_{\bar{\mathcal{K}}_{t-1}} z_t(x)\mathrm{d}x, \quad \delta_t := \log \frac{Z'_t}{Z_t}.$$

We first prove the following two claims:

(i) For all $t \in \{0, 1, \ldots, T\}$, we have $(1 - \gamma)\mathcal{K}^* + \gamma\bar{\mathcal{K}}_0 \subseteq \bar{\mathcal{K}}_t$.

(ii) For all $t \in [T]$, we have $\delta_t \geq \frac{s_t}{5}$.

We use induction to prove (i). It holds for $t = 0$ since $\mathcal{K}^* \subseteq \mathcal{K}_0 = \bar{\mathcal{K}}_0$ and $\mathcal{K}_0$ is convex. Suppose it holds for $t - 1$. If $\bar{\mathcal{K}}_t = \bar{\mathcal{K}}_{t-1}$, then it already holds for $t$. Otherwise, consider any separating hyperplane $(w, b) \in \mathbb{R}^d \times \mathbb{R}$ obtained in round $t$, which is the output of $\mathcal{SOD}(x', 2\gamma R, W)$ for some $x'$. By the guarantee of the SOD oracle, we have

$$w^\top x' \leq b - 2\gamma R \leq \min_{x^* \in \mathcal{K}^*} w^\top x^* - 2\gamma R.$$

This implies

$$(1 - \gamma)\mathcal{K}^* + \gamma\bar{\mathcal{K}}_0 \subseteq (1 - \gamma)\mathcal{K}^* + \gamma B(0, R) \subseteq \mathcal{K}^* + B(0, 2\gamma R) \subseteq \{x \in \mathbb{R}^d : w^\top x \geq b - 2\gamma R\}.$$

Note that $\{x \in \mathbb{R}^d : w^\top x \geq b - 2\gamma R\}$ is exactly the half-space to intersect with when updating $\mathcal{K}_t$. Hence we know that during the execution of the algorithm, $\mathcal{K}_t$ is always a superset of $(1-\gamma)\mathcal{K}^*+\gamma\bar{\mathcal{K}}_0$. This proves (i).

For (ii), note that each time $\mathcal{K}_t$ is updated, the mean of the distribution over $\mathcal{K}_t$ with density proportional to $z_t(x)$ is not included in the interior of the new $\mathcal{K}_t$. By Lemma D.2, this implies $\int_{\text{new } \mathcal{K}_t} z_t(x)\mathrm{d}x \leq \left(1 - \frac{1}{2e}\right) \int_{\text{old } \mathcal{K}_t} z_t(x)\mathrm{d}x$. Hence we have $-\delta_t = \log \frac{Z_t}{Z'_t} \leq \log \left(1 - \frac{1}{2e}\right)^{s_t}$, which gives $\delta_t \geq \frac{s_t}{5}$.

Now we show the regret bound. From (i) we know

$$(1 - \gamma)\mathcal{K}^* + \gamma\bar{\mathcal{K}}_0 \subseteq \bar{\mathcal{K}}_T \subseteq \bar{\mathcal{K}}_{T-1} \subseteq \cdots \subseteq \bar{\mathcal{K}}_1 \subseteq \bar{\mathcal{K}}_0 \subseteq B(0, R).$$

Therefore, we can apply Theorem B.3 to get

$$\forall x^* \in \mathcal{K}^* : \quad \sum_{t=1}^{T} (f_t(\bar{x}_t) - f_t(x^*)) \leq \frac{d \log \frac{1}{\gamma}}{\eta} + \eta L^2 R^2 T + \gamma LRT - \frac{\sum_{t=1}^{T} \delta_t}{\eta}$$

$$\leq \frac{d \log \frac{1}{\gamma}}{\eta} + \eta L^2 R^2 T + \gamma LRT - \frac{s}{5\eta},$$

where the second inequality is due to (ii).

The actual algorithm does not play $\bar{x}_t$, but a random $\tilde{x}_t$. Namely, letting $(V, p) \in \mathbb{R}^{d \times k} \times \Delta^{k-1}$ be the output of $\mathcal{SOD}(\bar{x}_t, 2\gamma R, W)$, we have that $\tilde{x}_t$ is equal to $v_i$ with probability $p_i$ $(i = 1, \ldots, k)$, where $V = (v_1, \ldots, v_k)$. By Theorem B.1 we know that there exists $c \in W^\circ$ such that $\|\sum_{i=1}^{k} p_i v_i + c - \bar{x}_t\| \leq 6\gamma R$, which implies (note $f_t \in W \cap B(0, L)$)

$$\mathbb{E}[f_t(\tilde{x}_t)] = f_t \left(\sum_{i=1}^{k} p_i v_i\right) \leq f_t \left(\sum_{i=1}^{k} p_i v_i + c\right) \leq f_t(\bar{x}_t) + 6\gamma LR.$$

Therefore we have

$$\forall x^* \in \mathcal{K}^* : \mathbb{E}\left[\sum_{t=1}^{T} (f_t(\tilde{x}_t) - f_t(x^*))\right] = \mathbb{E}\left[\sum_{t=1}^{T} (f_t(\tilde{x}_t) - f_t(\bar{x}_t))\right] + \sum_{t=1}^{T} (f_t(\bar{x}_t) - f_t(x^*))$$

$$\leq 6\gamma LRT + \frac{d \log \frac{1}{\gamma}}{\eta} + \eta L^2 R^2 T + \gamma LRT - \frac{s}{5\eta}$$

$$= \frac{d \log \frac{1}{\gamma}}{\eta} + \eta L^2 R^2 T + 7\gamma LRT - \frac{s}{5\eta}.$$

Setting $\gamma = \frac{1}{T}$ and $\eta = \frac{1}{LR} \min \left\{ 1, \sqrt{\frac{d \log T}{T}} \right\}$, the above bound becomes

$$\forall x^* \in \mathcal{K}^* : \mathbb{E} \left[ \sum_{t=1}^T (f_t(\tilde{x}_t) - f_t(x^*)) \right] \le LR \left( 7 + 2 \max \left\{ \sqrt{dT \log T}, d \log T \right\} \left( 1 - \frac{s}{10 d \log T} \right) \right).$$

We can use the above regret bound to bound the number of oracle calls in Algorithm 6. Since the regret is always lower bounded by $-2LRT$, the above regret upper bound implies $s = O(T)$. Therefore Algorithm 6 calls the SOD oracle for $s + T = O(T)$ times. Note that each implementation of SOD needs to call $\mathcal{O}_{\mathcal{K},\mathcal{K}^*}$ for $O\left( d \log \frac{4R}{2\gamma R} \right) = O(d \log T)$ times (Theorem B.1). We conclude that the total number of calls to $\mathcal{O}_{\mathcal{K},\mathcal{K}^*}$ in Algorithm 6 is at most $O(dT \log T)$. □

**Remark.** *Intuitively, when the SOD oracle is called in Algorithm 6, between the two outcomes (separation and decomposition) we should prefer decomposition, since this means we can make the play and move on to the next iteration. However, Theorem B.4 shows an interesting trade-off between oracle complexity and regret: the more oracle calls, the less the regret. This means obtaining separating hyperplanes helps the regret. Interestingly, we obtain our upper bound on the oracle calls by observing that regret can never be lower by $-2LRT$.*

## C   Proof for the Bandit Setting (Theorem 4.2)

We prove the following more general theorem than Theorem 4.2.

**Theorem C.1.** *Denote by $z_t$ the point played by Algorithm 3 in round $t$. Then for any $\gamma \in (0,1)$, $\epsilon \in (0, \alpha R]$ and $\eta > 0$, Algorithm 3 satisfies the following regret guarantee:*

$$\forall x^* \in \mathcal{K}: \quad \mathbb{E} \left[ \sum_{t=1}^T (f_t(z_t) - \alpha f_t(x^*)) \right] \le \frac{\alpha^2 \beta^2 d}{2\eta} + \frac{\eta L^2 R^2 d^2}{2\gamma} T + 2\gamma \alpha LRT + \epsilon LT,$$

*and the expected total number of calls to the oracle $\mathcal{O}_{\mathcal{K}, \alpha \mathcal{K}}$ in $T$ rounds is at most*

$$(1 + \gamma T) \left( 1 + 5d \log \frac{4\alpha R + 2R\sqrt{\alpha^2 \beta^2 d^2 + \frac{\eta^2 L^2 R^2 d^3}{\gamma} T + 6\eta \alpha LRdT}}{\epsilon} \right).$$

*In particular, setting $\eta = \frac{\alpha \beta^{4/3}}{LRT^{2/3}}$, $\epsilon = \frac{\alpha R}{T}$ and $\gamma = \frac{\beta^{2/3} d}{T^{1/3}}$ (assuming $T > \beta^2 d^3$ so $\gamma < 1$), we have*

$$\forall x^* \in \mathcal{K}: \quad \mathbb{E} \left[ \sum_{t=1}^T (f_t(z_t) - \alpha f_t(x^*)) \right] \le \alpha LR \left( 3d(\beta T)^{2/3} + 1 \right),$$

*and the expected total number of oracle calls in $T$ rounds is at most $O\left( d^2 (\beta T)^{2/3} \log T \right)$.*

*Proof.* Let $x_t$, $y_t$ and $\tilde{x}_t$ be the same $x_t$, $y_t$ and $\tilde{x}_t$ appearing in Algorithm 1 during our implementation. We define $\bar{x}_t := \mathbb{E}[\tilde{x}_t | y_t]$ similarly to the proof of Theorem 3.1. It is easy to see that $(\varphi, W')$ satisfies the PNIP property (Definition 2.4) and $\tilde{f}_t \in W'$ for all $t \in [T]$, where $W' = (M^\top)^{-1} \mathbb{R}_+^d$.

Note that $\nabla \varphi(x) = Q^{-1} x$, which implies

$$D_\varphi(x_t, y_{t+1}) = \frac{1}{2} (x_t - y_{t+1})^\top Q^{-1} (x_t - y_{t+1}) = \frac{1}{2} (\nabla \varphi(x_t) - \nabla \varphi(y_{t+1}))^\top Q (\nabla \varphi(x_t) - \nabla \varphi(y_{t+1}))$$

$$= \frac{\eta^2}{2} \tilde{f}_t^\top Q \tilde{f}_t.$$

Using the regret bound (3) in the proof of Theorem 3.1, for any $x^* \in \mathcal{K}$ we have

$$\sum_{t=1}^{T} \left( \tilde{f}_t(x_t) - \alpha \tilde{f}_t(x^*) \right) \leq \frac{1}{\eta} \left( D_\varphi(\alpha x^*, y_1) - D_\varphi(\alpha x^*, y_{T+1}) + \sum_{t=1}^{T} D_\varphi(x_t, y_{t+1}) \right)$$

$$\leq \frac{1}{\eta} \left( \varphi(\alpha x^*) - \min_{y \in \mathbb{R}^d} \varphi(y) + \sum_{t=1}^{T} D_\varphi(x_t, y_{t+1}) \right) \tag{7}$$

$$= \frac{1}{\eta} \varphi(\alpha x^*) + \frac{\eta}{2} \sum_{t=1}^{T} \tilde{f}_t^\top Q \tilde{f}_t.$$

Since $\{q_i\}_{i=1}^{d}$ is a $\beta$-BS($\mathcal{K}$), there exist $\beta_1, \ldots, \beta_d \in [-\beta, \beta]$ such that $x^* = \sum_{i=1}^{d} \beta_i q_i$. Then we have

$$\varphi(\alpha x^*) = \frac{1}{2}(\alpha x^*)^\top Q^{-1}(\alpha x^*) = \frac{\alpha^2}{2} \|M^{-1}x^*\|^2 = \frac{\alpha^2}{2} \left\| \sum_{i=1}^{d} \beta_i M^{-1} q_i \right\|^2 = \frac{\alpha^2}{2} \left\| \sum_{i=1}^{d} \beta_i e_i \right\|^2 \leq \frac{\alpha^2 \beta^2 d}{2}.$$

We also have

$$\mathbb{E}\left[ \tilde{f}_t^\top Q \tilde{f}_t \right] = \gamma \sum_{i=1}^{d} \frac{1}{d} \left( \frac{d}{\gamma} q_i^\top f_t \right)^2 q_i^\top Q^{-1} Q Q^{-1} q_i = \frac{d}{\gamma} \sum_{i=1}^{d} \left( q_i^\top f_t \right)^2 q_i^\top Q^{-1} q_i$$

$$\leq \frac{L^2 R^2 d}{\gamma} \sum_{i=1}^{d} q_i^\top (MM^\top)^{-1} q_i = \frac{L^2 R^2 d}{\gamma} \sum_{i=1}^{d} e_i^\top e_i = \frac{L^2 R^2 d^2}{\gamma}.$$

Hence by taking expectation on (7) we get

$$\mathbb{E}\left[ \sum_{t=1}^{T} \left( \tilde{f}_t(x_t) - \alpha \tilde{f}_t(x^*) \right) \right] \leq \frac{\alpha^2 \beta^2 d}{2\eta} + \frac{\eta L^2 R^2 d^2}{2\gamma} T. \tag{8}$$

Note that $\mathbb{E}[\tilde{f}_t | x_t] = \gamma \sum_{i=1}^{d} \frac{1}{d} \frac{d}{\gamma} Q^{-1} q_i q_i^\top f_t = Q^{-1} \left( \sum_{i=1}^{d} q_i q_i^\top \right) f_t = f_t$. Therefore (8) becomes

$$\mathbb{E}\left[ \sum_{t=1}^{T} (f_t(x_t) - \alpha f_t(x^*)) \right] \leq \frac{\alpha^2 \beta^2 d}{2\eta} + \frac{\eta L^2 R^2 d^2}{2\gamma} T. \tag{9}$$

Next, by the guarantee of the PAD oracle, for any $t \in [T]$ we know that there exists $c_t \in (W')^\circ$ such that $\|\overline{x}_t + c_t - x_t\| \leq \epsilon$. It is easy to see that $\mathbb{R}_+^d \subseteq W'$, which implies $(W')^\circ \subseteq \mathbb{R}_+^d$, so we know $c_t \in \mathbb{R}_+^d$. Then we have

$$\forall t \in [T]: \quad \mathbb{E}\left[ f_t(\overline{x}_t) - f_t(x_t) \right] = \mathbb{E}\left[ f_t^\top (\overline{x}_t - x_t) \right] \leq \mathbb{E}\left[ f_t^\top (\overline{x}_t + c_t - x_t) \right] \leq \epsilon L.$$

Thus (9) implies

$$\mathbb{E}\left[ \sum_{t=1}^{T} (f_t(\overline{x}_t) - \alpha f_t(x^*)) \right] \leq \frac{\alpha^2 \beta^2 d}{2\eta} + \frac{\eta L^2 R^2 d^2}{2\gamma} T + \epsilon L T. \tag{10}$$

Finally, since the point played in round $t$, $z_t$, is equal to $\tilde{x}_t$ (whose expectation is $\overline{x}_t$) with probability $1 - \gamma$, we have

$$\mathbb{E}\left[ \sum_{t=1}^{T} (f_t(z_t) - \alpha f_t(x^*)) \right] \leq (1 - \gamma) \mathbb{E}\left[ \sum_{t=1}^{T} (f_t(\tilde{x}_t) - \alpha f_t(x^*)) \right] + \gamma \cdot 2\alpha LRT$$

$$= (1 - \gamma) \mathbb{E}\left[ \sum_{t=1}^{T} (f_t(\overline{x}_t) - \alpha f_t(x^*)) \right] + 2\gamma \alpha LRT$$

$$\leq \frac{\alpha^2 \beta^2 d}{2\eta} + \frac{\eta L^2 R^2 d^2}{2\gamma} T + \epsilon L T + 2\gamma \alpha LRT.$$

**Oracle complexity.** Using Theorem 3.4, we know that when $b_t = \text{EXPLORE}$, the number of calls to the oracle $\mathcal{O}_{\mathcal{K},\alpha\mathcal{K}}$ in round $t$ is at most $\left\lceil 5d \log \frac{4\alpha R + 2\sqrt{\frac{2}{\mu} \min_{x^* \in \mathcal{K}} D_\varphi(\alpha x^*, y_t)}}{\epsilon} \right\rceil$, where $\mu$ is the strong convexity parameter of $\varphi$.

In the above proof of the regret bound, we have ignored the term $D_\varphi(\alpha x^*, y_{T+1})$ in (7). If we instead keep this term, the regret bound (10) will become

$$\forall x^* \in \mathcal{K}: \quad \mathbb{E}\left[ \sum_{t=1}^{T} (f_t(\overline{x}_t) - \alpha f_t(x^*)) \right] \leq \frac{\alpha^2 \beta^2 d}{2\eta} + \frac{\eta L^2 R^2 d^2}{2\gamma} T + \epsilon L T - \frac{1}{\eta} \mathbb{E}\left[ D_\varphi(\alpha x^*, y_{T+1}) \right].$$

In the above inequality, substituting $T$ with $t$, we have

$$\forall t \in [T], \forall x^* \in \mathcal{K}: \quad \mathbb{E}\left[ D_\varphi(\alpha x^*, y_{t+1}) \right] \leq \frac{\alpha^2 \beta^2 d}{2} + \frac{\eta^2 L^2 R^2 d^2}{2\gamma} t + \epsilon \eta L t - \eta \mathbb{E}\left[ \sum_{j=1}^{t} (f_j(\overline{x}_j) - \alpha f_j(x^*)) \right]$$

$$\leq \frac{\alpha^2 \beta^2 d}{2} + \frac{\eta^2 L^2 R^2 d^2}{2\gamma} T + \epsilon \eta L T + \eta \cdot 2\alpha L R T$$

$$\leq \frac{\alpha^2 \beta^2 d}{2} + \frac{\eta^2 L^2 R^2 d^2}{2\gamma} T + 3\eta \alpha L R T.$$

The above upper bound is also clearly valid for $D_\varphi(\alpha x^*, y_1)$.

Since $\varphi(x) = \frac{1}{2} x^\top Q^{-1} x$ is quadratic, we know that $\mu = \lambda_{\min}(Q^{-1}) = \frac{1}{\lambda_{\max}(Q)} = \frac{1}{\max_{u \in \mathbb{R}^d, \|u\|=1} \|Qu\|} = \frac{1}{\max_{u \in \mathbb{R}^d, \|u\|=1} \|\sum_{i=1}^{d} q_i q_i^\top u\|} \geq \frac{1}{\sum_{i=1}^{d} \|q_i\|^2} \geq \frac{1}{R^2 d}$, where $\lambda_{\min}(P)$ and $\lambda_{\max}(P)$ are respectively the smallest and the largest eigenvalues of a symmetric matrix $P$.

Note that $\log(a + \sqrt{x})$ is a concave function in $x$ for $a > 0$. By Jensen's inequality, the expected number of calls to the oracle $\mathcal{O}_{\mathcal{K}}^\alpha$ in round $t$ when $b_t = \text{EXPLORE}$ is upper bounded by:

$$\mathbb{E}\left[ 1 + 5d \log \frac{4\alpha R + 2\sqrt{\frac{2}{\mu} \min_{x^* \in \mathcal{K}} D_\varphi(\alpha x^*, y_t)}}{\epsilon} \right]$$

$$\leq 1 + \min_{x^* \in \mathcal{K}} \mathbb{E}\left[ 5d \log \frac{4\alpha R + 2\sqrt{\frac{2}{\mu} D_\varphi(\alpha x^*, y_t)}}{\epsilon} \right]$$

$$\leq 1 + \min_{x^* \in \mathcal{K}} 5d \log \frac{4\alpha R + 2\sqrt{\frac{2}{\mu} \mathbb{E}\left[ D_\varphi(\alpha x^*, y_t) \right]}}{\epsilon}$$

$$\leq 1 + 5d \log \frac{4\alpha R + 2\sqrt{2 R^2 d \left( \frac{\alpha^2 \beta^2 d}{2} + \frac{\eta^2 L^2 R^2 d^2}{2\gamma} T + 3\eta \alpha L R T \right)}}{\epsilon}$$

$$= 1 + 5d \log \frac{4\alpha R + 2R\sqrt{\alpha^2 \beta^2 d^2 + \frac{\eta^2 L^2 R^2 d^3}{\gamma} T + 6\eta \alpha L R d T}}{\epsilon}.$$

Therefore the expected total number of calls to the oracle $\mathcal{O}_{\mathcal{K}}^\alpha$ in $T$ rounds is at most

$$(1 + \gamma T) \left( 1 + 5d \log \frac{4\alpha R + 2R\sqrt{\alpha^2 \beta^2 d^2 + \frac{\eta^2 L^2 R^2 d^3}{\gamma} T + 6\eta \alpha L R d T}}{\epsilon} \right).$$

The second part of the theorem can be directly verified using the specific choices of $\eta$, $\epsilon$ and $\gamma$ and noting $\log(\text{poly}(\beta d)) = O(\log T)$ since $T > \beta^2 d^3$. $\qquad\square$

# D  Log-Concave Distributions

A distribution over $\mathbb{R}^d$ with a density function $f$ is *log-concave* if $\log(f)$ is a concave function. For a convex set $\mathcal{S}$ equipped with a membership oracle, there exist polynomial-time algorithms for

sampling from any log-concave distribution over $\mathcal{S}$ (Lovász and Vempala, 2007). This can be used to approximately compute the mean of any log-concave distribution.

We have the following classical result which says that every half-space close enough to the mean of a log-concave distribution must contain at least constant probability mass. For simplicity, we only state and prove the result for isotropic (i.e., identity covariance) log-concave distributions, but the result can be easily generalized to allow arbitrary covariance.

**Lemma D.1.** *Consider any isotropic (identity covariance) log-concave distribution $p$ over $\mathbb{R}^d$ with mean $x^*$. Then for any half-space $H$ such that $\|x^* - \Pi_H(x^*)\| \leq \frac{1}{2e}$, we have $\int_H p(x)\mathrm{d}x \geq \frac{1}{2e}$.*

*Proof.* Let $H = \{x \in \mathbb{R}^d : w^\top x \geq b\}$ for a unit vector $w \in \mathbb{R}^d$ and $b \in \mathbb{R}$, and assume without loss of generality that $x^* = 0$. Consider the one-dimensional random variable $Y := w^\top X - b$, where $X \sim p$. Denote by $q : \mathbb{R} \to \mathbb{R}$ the density function of $y$. Then we have

$$\int_H p(x)\mathrm{d}x = \int_0^\infty q(y)\mathrm{d}y.$$

Let $y^* := \mathbb{E}[Y] = w^\top x^* - b = -b$. By our assumption, we know $|y^*| \leq \frac{1}{2e}$. Moreover, since log-concavity is preserved under linear transformations (Prékopa, 1973), we know that $y$ also follows a log-concave distribution, and it is easy to see that it is also isotropic. Using Lemma 5.4 in (Lovász and Vempala, 2007), we know $\int_{y^*}^\infty q(y)\mathrm{d}y \geq \frac{1}{e}$. In addition, from Lemma 5.5 in (Lovász and Vempala, 2007) we know $q(y) \leq 1$ ($\forall y \in \mathbb{R}$). Therefore, we have

$$\frac{1}{e} - \int_0^\infty q(y)\mathrm{d}y \leq \int_{y^*}^\infty q(y)\mathrm{d}y - \int_0^\infty q(y)\mathrm{d}y \leq |y^*| \sup_{y \in \mathbb{R}} q(y) \leq \frac{1}{2e},$$

which implies $\int_0^\infty q(y)\mathrm{d}y \geq \frac{1}{2e}$, completing the proof. $\square$

As an implication, we have the following lemma regarding mean computation of a log-concave distribution, which is useful in this paper.

**Lemma D.2.** *For any log-concave distribution $p$ in $\mathbb{R}^d$ with mean $x^*$, whose support $\mathrm{supp}(p)$ is in $B(0, R)$ ($R > 0$), and any $\epsilon > 0$ and $0 < \delta < 1$, it is possible to compute a point $\tilde{x}^*$ in $\mathrm{poly}\left(d, \frac{1}{\epsilon}, \log\frac{1}{\delta}\right)$ time such that with probability at least $1 - \delta$ we have: (1) $\|\tilde{x}^* - x^*\| \leq R\epsilon$; (2)for any half space $H$ containing $\tilde{x}^*$, $\int_H p(x)\mathrm{d}x \geq \frac{1}{2e}$.*

For our purpose in this paper, it always suffices to choose $\epsilon = \frac{1}{T}$ and $\delta = \frac{1}{\mathrm{poly}(T)}$ ($T$ being the total number of rounds) without hurting our regret bounds.