[Reviews · NeurIPS 2018]

Reviewer 1



The authors consider the problem of online linear optimization through an access to an approximation oracle. The paper focuses on decreasing the number of oracle access per round while providing the same alpha-regret as the state-of-the-art. Specifically, the authors show that under full information Algorithm 1 attains O(sqrt(T)) alpha-regret with O(log(T)) oracle calls per round. Under bandit feedback Algorithm 3 attains O(T^{2/3}) alpha-regret with total O(T^{2/3} log(T)) calls in T rounds. The key innovation is the use of the online mirror descent algorithm for online learning aided with the oracle calls, which enables the improvements in terms of oracle calls. Pros: * The paper significantly improves upon the best known results in online linear optimization with access to approximation oracles. Both under full information and bandit feedback the proposed algorithms improve upon the number of oracle access per round. * The results on full feedback consider a more general oracle than alpha-approximation oracle. * The exposition is precise and the writing is fluent. The Algorithm 1 and Algorithm 2 are easy to follow. The proofs are novel and clearly presented in the supplementary material. Cons: * In Table 1, the authors provide $\tilde{O}(1)$ as the bounds on the number of calls to the oracle on each round. It is not appropriate as $\tilde$ notation is used when hidden factors are smaller than the one presented. The authors may use $O(log(T))$ as the bound * The paper is theoretical and lacks any experimental validation or comparison. * The continuous multiplicative weight update algorithm and its guarantees are completely deferred to Appendix B in the supplementary material. This makes the results inaccessible from the main paper. * The importance of the polynomial time sampling from a log-concave distribution in the proposed algorithm is completely omitted from the main body of the paper. Further, in Appendix D the connection of such sampling to the proposed algorithms is not clearly written. * The geometric intuition of Lemma 3.8 needs more elaboration.

Reviewer 2



In the setting of online combinatorial optimization (OCO) with linear approximation oracles, this paper provides a novel approach for achieving a low regret, while limiting the number of calls to the oracle. Specifically, the authors examine a new class of OCO problems, where the domains of the convex regularization function and the linear loss functions satisfy the pairwise non-negative inner product property (PNIP). For this class, the authors provide algorithms in both the full information case and the bandit case using an approximate, yet controllable, projection-decomposition operation. In both case, the expected per-round number of calls to the linear approximation oracle is essentially logarithmic in the horizon T. Overall I have a mixed opinion about this paper. On the one hand, the authors are investigating an interesting class of OCO problems characterized by the PNIP property. Since many regularizers in the ML literature satisfy this property, this approach is clearly interesting from a theoretical viewpoint. Furthermore, the regret analysis derived for this class of problems is original, and hence, my tendency is to lean toward accepting this paper. On the other hand, I have several concerns about the computational complexity of the algorithms. As justified by the first sentence of the introduction, the key motivation of the authors is to “efficiently” learn a given problem using optimization oracles. The main contribution of this paper is to restrict the number of calls to the approximation oracle, while maintaining a low regret. This is fine, but nothing is said about the “overall” per-round runtime complexity of the algorithms. My point is that many, if not most, linear approximation oracles run in low polynomial time. For example, if we focus on MAXCUT with SDP approximation, then Klein and Liu (STOC’96) provide an algorithm that runs in time \tilde O(mn), and Kale (PhD Th’07) provides a faster method that runs in time \tilde O(m) for certain classes of graphs. This is in stark contrast with Bregman projection steps (Line 4 of Algorithm 2) and volume calculations (Line 3 of Algorithm 2) for which the runtime complexity can generally be much greater. It seems that the specific structures of the domain of D_{\phi} and the polyhedron W_{i} in Line 4 of Algorithm 2 might allow faster techniques for these operations. But this should be explained in more detail, and the runtime complexity of Algorithm 2 should be clarified in order to highlight the practical feasibility of the method. Minor comments: * For the sake of clarity, it would be nice to use the same metrics for the third column of Table 1 and Table 2 (ex: use the per-round number of calls to the oracle). * In the paper, the notion of strong convexity is defined for the Euclidean norm. I am wondering whether better results for Theorem 1.1 could be obtained using different norms (ex: as we know, the un-normalized relative entropy in Line 125 is strongly convex with respect to || ||_1, which yields a logarithmic dependence on the input dimension using online mirror descent). * Line 224: “combining” Lemmas 3.7 and 3.8.

Reviewer 3



This paper mainly considered the online linear optimization problem with approximation oracle (as well as its bandit version), which had been studied for a long time. The problem is important in real world especially when we can only solve the underlying offline optimization problems approximately, such as max-cut etc. The main contribution of this paper is to propose several algorithms which are proved to be more efficient (in terms of oracle calls per round) than previous state-of-the-art algorithm while achieve nearly the same regret guarantees. In detail, authors improved previous O(\sqrt{T}) oracle complexity per round to O(log T) order (both achieve O(\sqrt{T}) regret) in full information setting, and improved previous O(T) oracle complexity per round to O(T^{2/3}*log T) order (both achieve O(T^{2/3}) regret) in bandit setting. The paper is well written and structured in a clean way. Some techniques look interesting, especially the projection and decomposition oracle. Pros: 1. Two algorithms based on different frameworks (i.e. OMD and continual exponential weights) are proposed in full information setting. Both of them are more efficient than previous methods. OMD based algorithm looks more succinct while only holds for loss functions with PNIP property. Continual exponential weights based algorithm works for general loss functions. 2. An algorithm is designed for bandit setting based on above OMD type algorithm with a novel regularization function, which is also more efficient than previous algorithms. Cons: 1. A critical condition assumed in this paper is that the mean of any log-concave distribution can be exactly calculated, and all of the proposed algorithms heavily rely on calculating the gravity center of underlying constraint set. However, in practical applications, we need to approximate this gravity center, and it will cause some approximation error, which will demonstrate a trade-off between computational efficiency and final regret performance and it is exactly my main concern about this paper. According to the explanation in appendix D, this approximation error \epsilon is set as 1/T, then calculating the approximate gravity center will cost time poly(1/\epsilon) = poly(T) per round, which degenerates the computational performance of proposed algorithms. If we want to guarantee the efficiency of approximate mean calculation algorithm, then \epsilon should be set at least in 1/\log(T) order, then I am not sure whether this approximation error is enough to guarantee the final O(\sqrt{T}) regret performance; 2. Could authors explain why OMD based algorithm can achieve better efficiency than the algorithm proposed in Garber 2017 paper? I know his algorithm is based on Online Gradient Descent and ellipsoid method. Maybe the critical part is the projection and decomposition oracle constructed in this paper. It will be better if authors could provide some intuition about this oracle before stating four detailed lemmas. Other minor comments: 1. In table 1, it will be better to add a notation about alg 6, as it only appears in the appendix; 2. In the equation under line 307 in appendix, it should be f(x) – f(y) <= \frac{1}{2\mu} … Update: Thanks for authors' response. Though the main focus of this paper is oracle complexity, but I think computational complexity is also a critical part. Since authors mentioned using ellipsoid method could lead to a better computational complexity, which solved my main concern, I hope to see the corresponding part in the further version.